# Private Stochastic Optimization for Achieving Second-Order Stationary Points

## Abstract

This paper addresses the challenge of achieving second-order stationary points (SOSP) in differentially private stochastic non-convex optimization. We identify two key limitations in the state-of-the-art: (i) inaccurate error rates caused by the omission of gradient variance in saddle point escape analysis, resulting in inappropriate parameter choices and overly optimistic performance estimates, and (ii) inefficiencies in private SOSP selection via the AboveThreshold algorithm, particularly in distributed learning settings, where perturbing and sharing Hessian matrices introduces significant additional noise. To overcome these challenges, we revisit perturbed stochastic gradient descent (SGD) with Gaussian noise and propose a new framework that leverages general gradient oracles. This framework introduces a novel criterion based on model drift distance, ensuring provable saddle point escape and efficient convergence to approximate local minima with low iteration complexity. Using an adaptive SPIDER as the gradient oracle, we establish a new DP algorithm that corrects existing error rates. Furthermore, we extend our approach to a distributed adaptive SPIDER, applying our framework to distributed learning scenarios and providing the first theoretical results on achieving SOSP under differential privacy in distributed environments with heterogeneous data. Finally, we analyze the limitations of the AboveThreshold algorithm for private model selection in distributed learning and show that as model dimensions increase, the selection process introduces additional errors, further demonstrating the superiority of our proposed framework.

## 1 Introduction

Stochastic optimization is one of the most fundamental problems in machine learning and statistics, with the goal of building models that generalize well to unseen data using only a limited number of i.i.d. samples drawn from an unknown distribution. As the volume of sensitive data grows, ensuring privacy during the training process has become a critical concern. This has led to the adoption of differential privacy (DP) (Dwork et al., 2006) in stochastic optimization, which provides strong privacy guarantees while preserving the utility of the learned model.

Over the past decade, significant advances have been made in DP stochastic optimization, particularly for convex objectives, e.g., (Choquette-Choo et al., 2024; Liu & Asi, 2024; Su et al., 2023; 2022; Tao et al., 2022). While convex optimization is relatively well-understood, the non-convex setting presents additional challenges due to the existence of saddle points. In non-convex optimization, most existing DP algorithms focus on achieving convergence to first-order stationary points (FOSP), where the gradient norm is small (Arora et al., 2023; Bassily et al., 2021; Zhou et al., 2020). However, this criterion is often insufficient, as FOSP can include both local maxima and saddle points—where saddle points represent highly sub-optimal solutions in many problems, as shown by Jain et al. (2015) and Sun et al. (2016). For practical non-convex functions, second-order stationary points (SOSP)—where the gradient is small and the Hessian is positive semi-definite—are preferred, as they guarantee convergence to a local minimum.

Due to the importance of achieving SOSP, substantial effort has been devoted to this area, as demonstrated by works such as Fang et al. (2019), Jin et al. (2021), Daneshmand et al. (2018), Jin et al. (2017), and Ge et al. (2015). However, few algorithms specifically target this more stringent criterion under the DP framework. The notable exception and current state-of-the-art for achieving SOSP

under DP is the work of Liu et al. (2024), which proposes adding Gaussian noise, scaled according to the gradient error, to ensure that the DP-SGD sequence escapes each potential saddle point. Despite being the state-of-the-art, we identify significant gaps in their utility analysis and error rate guarantees. Specifically, the omission of gradient variance in their saddle point escape analysis results in incorrect parameter settings and overly optimistic performance estimates (see Section 3).

Beyond these challenges, distributed learning serves as another key motivation for this work, becoming increasingly important in modern machine learning, particularly with the rise of large-scale models that require decentralized systems for efficient training. To date, no existing work has explored DP non-convex stochastic optimization in distributed settings with the goal of achieving SOSP. Distributed learning introduces additional challenges, such as data heterogeneity and the need for privacy-preserving protocols across multiple clients. Moreover, the current state-of-the-art approach by Liu et al. (2024) suffers from severe performance degradation when applied to distributed settings. Their reliance on the AboveThreshold algorithm for private model selection introduces significant noise when sharing perturbed Hessian matrices across clients, particularly in high-dimensional scenarios (see detailed discussion in Section 6). This degradation can be attributed to their learning algorithm, which only guarantees the existence of an SOSP among all model iterates during the learning process and thus requires additional use of private model selection algorithms.

**Our Contributions.** To address the gaps mentioned above, we propose a new algorithmic framework and analysis for DP stochastic non-convex optimization that ensures convergence to SOSP. Our contributions can be summarized as follows:

**1. Revisiting Non-Convex Stochastic Optimization Beyond DP:** We propose a new perturbed stochastic gradient descent (PSGD) framework with Gaussian noise for perturbation, utilizing general stochastic gradient oracles. This framework serves as a general optimization tool for non-convex stochastic optimization, applicable beyond the context of DP. In this framework, we introduce a novel criterion based on model drift distance to determine SOSP, ensuring provable escape from saddle points and efficient convergence to approximate local minima with low iteration complexity and high probability.

**2. Corrected Error Rates for DP Non-Convex Optimization:** By employing an adaptive DP-SPIDER as the perturbed gradient oracle, we establish corrected error rates for achieving SOSP under DP in non-convex optimization. Specifically, we adjust the previous state-of-the-art error rate from $\tilde{O}\left(\frac{1}{n^{\frac{1}{3}}} + \left(\frac{\sqrt{d}}{\epsilon n}\right)^{\frac{3}{7}}\right)$ to $\tilde{O}\left(\frac{1}{n^{\frac{1}{3}}} + \left(\frac{\sqrt{d}}{\epsilon n}\right)^{\frac{2}{5}}\right)$.

**3. Application to Distributed Learning:** We extend the adaptive DP-SPIDER framework to distributed settings. With this new estimator, our algorithm offers an adaptive improvement over the previous DIFF2 algorithm Murata & Suzuki (2023), which only guarantees convergence to FOSP under DP. Our approach provides the first DP error rate for attaining SOSP in distributed learning with heterogeneous data. Additionally, we analyze the limitations of the AboveThreshold algorithm for private model selection in distributed learning scenarios, particularly in high-dimensional settings. We show that this selection process degrades the error rate guaranteed by the learning algorithm, highlighting the superiority of our proposed framework.

Due to the space limit, the literature review, technical preliminaries, along with all omitted proofs are included in the Appendix.

## 2 PRELIMINARIES

**Notations** We use $\|\cdot\|$ to denote the $\ell_2$ norm and $\lambda_{\min}(\cdot)$ to represent the smallest eigenvalue of a matrix. The notation $\mathbf{I}_d$ denotes the $d$-dimensional identity matrix. We use $O(\cdot)$ and $\Omega(\cdot)$ to hide constants independent of problem parameters, while $\tilde{O}(\cdot)$ and $\tilde{\Omega}(\cdot)$ additionally hide factors that depend only polylogarithmically on the problem parameters.

**Stochastic Optimization** Let $f : \mathbb{R}^d \times \mathcal{Z} \to \mathbb{R}$ be a (potentially non-convex) loss function, where the input consists of the $d$-dimensional model parameter $x \in \mathbb{R}^d$ and a data point $z \in \mathcal{Z}$.

**Assumption 1.** We assume that $f(\cdot; z)$ is $G$-Lipschitz, $M$-smooth, and $\rho$-Hessian Lipschitz. Specifically, for any $z \in \mathcal{Z}$ and any $x_1, x_2 \in \mathbb{R}^d$, we have:

- $|f(x_1; z) - f(x_2; z)| \leq G\|x_1 - x_2\|$,

- $\|\nabla f(x_1; z) - \nabla f(x_2; z)\| \leq M\|x_1 - x_2\|$,

- $\|\nabla^2 f(x_1; z) - \nabla^2 f(x_2; z)\| \leq \rho\|x_1 - x_2\|$.

For a dataset $Z \subseteq \mathcal{Z}$, we define its empirical risk function as $f(x; Z) := \frac{1}{|Z|} \sum_{z \in Z} f(x; z)$. If the data points are sampled i.i.d. from an unknown distribution $\mathcal{D}$, the population risk function of a model $x$, denoted $F(x; \mathcal{D})$, is defined as $F(x; \mathcal{D}) := \mathbb{E}_{z \sim \mathcal{D}}[f(x; z)]$. For simplicity, we refer to the population risk as $F(x)$ when the distribution $\mathcal{D}$ is clear from context.

**Assumption 2.** Let $x^*$ represent the population risk minimizer and $F^*$ the corresponding minimum risk. We assume $\max_x F(x) - F^* \leq U$ for some upper bound $U$.

Given a dataset $D$ with $n$ i.i.d. samples drawn from $\mathcal{D}$, our goal is to find an $\alpha$-second-order stationary point ($\alpha$-SOSP).

**Definition 1** ($\alpha$-SOSP). An $\alpha$-SOSP $x$ of the population risk $F(\cdot)$ satisfies $\|\nabla F(x)\| \leq \alpha$ and $\nabla^2 F(x) \succeq -\sqrt{\rho\alpha} \cdot \mathbf{I}_d$.

The notion of $\alpha$-SOSP excludes $\alpha$-strict saddle points where $\nabla^2 F(x) \preceq -\sqrt{\rho\alpha} \cdot \mathbf{I}_d$, ensuring convergence to an approximate local minimum (with local maxima considered a special case of saddle points). Following prior works on finding $\alpha$-SOSP such as (Liu et al., 2024; Jin et al., 2021), we assume $M \geq \sqrt{\rho\alpha}$ to ensure that finding a second-order stationary point is strictly more challenging than finding a first-order stationary point.

**Distributed Learning** In the distributed (federated) learning setting, $m$ clients collaboratively learn under the coordination of a central server. Each client $j \in [m]$ has a local dataset $D_j$ of size $n$, sampled from an unknown local distribution $\mathcal{D}_j$. The population risk for client $j$ is defined as $F(x; \mathcal{D}_j) := \mathbb{E}_{z \sim \mathcal{D}_j}[f(x; z)]$. For brevity, we refer to the population risk of client $j$ as $F_j(x)$. In the distributed setting, the global population risk for any model $x$, denoted $F(x; \mathcal{D})$ or simply $F(x)$, is defined as $F(x) := \frac{1}{m} \sum_{j \in [m]} F_j(x)$. We allow for heterogeneity in the local datasets, meaning that the local distributions $\{\mathcal{D}_j\}_{j \in [m]}$ may differ arbitrarily.

**Differential Privacy** We aim to achieve SOSP while ensuring privacy under the framework of Differential Privacy (DP). Two datasets $D$ and $D'$ are called *adjacent* if they differ by at most one record. DP ensures that the output of a learning algorithm on any pair of adjacent datasets is statistically indistinguishable.

**Definition 2** (Differential Privacy (DP) (Dwork et al., 2006)). Given $\epsilon, \delta > 0$, a randomized algorithm $\mathcal{A} : \mathcal{Z} \to \mathcal{X}$ is $(\epsilon, \delta)$-DP if for any pair of adjacent datasets $D, D' \subseteq \mathcal{Z}$, and any measurable subset $S \subseteq \mathcal{X}$,

$$\mathbb{P}[\mathcal{A}(D) \in S] \leq \exp(\epsilon) \cdot \mathbb{P}[\mathcal{A}(D') \in S] + \delta. \tag{1}$$

In distributed learning, we focus on inter-client record-level DP (ICRL-DP), which assumes that clients do not trust the server or other clients with their sensitive local data. This notion has been widely adopted in state-of-the-art distributed learning works, such as Gao et al. (2024); Lowy et al. (2023); Lowy & Razaviyayn (2023).

**Definition 3** (Inter-Client Record-Level DP (ICRL-DP)). Given $\epsilon, \delta > 0$, a randomized algorithm $\mathcal{A} : \mathcal{Z}^m \to \mathcal{X}$ satisfies $(\epsilon, \delta)$-ICRL-DP if, for any client $j \in [m]$ and any pair of local datasets $D_j$ and $D'_j$, the full transcript of client $j$'s sent messages during the learning process satisfies (1), assuming fixed local datasets for other clients.

**Variance Reduction via SPIDER** In standard SGD and its variants, a gradient estimate $g_t$ is used at each iteration $t$ to approximate the true gradient $\nabla F(x_{t-1})$. However, stochastic gradients computed from batches or individual samples often exhibit high variance, which can degrade learning performance. The Stochastic Path Integrated Differential Estimator (SPIDER), introduced by Fang et al. (2018), addresses this issue by using two gradient oracles, $\mathcal{O}_1$ and $\mathcal{O}_2$, to reduce variance, given a batch of data samples $\mathcal{B}_t$ at each iteration $t$:

- Oracle $\mathcal{O}_1(x_{t-1}, \mathcal{B}_t) := \nabla f(x_{t-1}; \mathcal{B}_t)$ provides an estimate of $\nabla F(x_{t-1})$.
- Oracle $\mathcal{O}_2(x_{t-1}, x_{t-2}, \mathcal{B}_t) := \nabla f(x_{t-1}; \mathcal{B}_t) - \nabla f(x_{t-2}; \mathcal{B}_t)$ approximates the gradient difference $\nabla F(x_{t-1}) - \nabla F(x_{t-2})$.

SPIDER periodically queries $\mathcal{O}_1$ every $l$ iterations for an updated gradient estimate $g_t$. In the remaining $l - 1$ iterations, it uses $\mathcal{O}_2$ to estimate the gradient difference and updates the gradient estimate as $g_t = g_{t-1} + \mathcal{O}_2(x_{t-1}, x_{t-2}, \mathcal{B}_t)$. For smooth functions, the variance of the estimate $\nabla F(x_{t-1}) - \nabla F(x_{t-2})$ is proportional to $\|x_{t-1} - x_{t-2}\|$, which is typically small when the model drift between iterations is minimal. This enables SPIDER to effectively reduce gradient variance while maintaining high accuracy in gradient estimation.

## 3 GAPS AND LIMITATIONS IN STATE-OF-THE-ART

**Gaps in Error Rate Analysis**  The state-of-the-art error rate for achieving an $\alpha$-SOSP in population risk function under DP, as presented in Liu et al. (2024), contains fundamental gaps that lead to incorrect conclusions. First, the error analysis is based on Lemma 3.4, which is essentially derived from Wang et al. (2019, Lemma 12). This lemma asserts that adding Gaussian noise at the same scale as the gradient estimation error can sufficiently reduce the function value with high probability, ensuring successful escape from saddle points. The key to the proof lies in demonstrating that the region around the saddle point, where SGD may get stuck, is narrow. This ensures that at least one of two coupled SGD sequences, initialized a certain distance apart in the escape direction due to the perturbation, can successfully escape.

However, the existing analysis overlooks a critical factor: the stochastic gradient variance. In analyzing the dynamics of the coupled points, the authors used exact gradients of the population risk, as evidenced in the equation preceding equation (39) of Wang et al. (2019). This oversight leads to incorrect parameter settings, particularly with respect to the step size $\eta$. Even for non-private SGD, Jin et al. (2021) has shown that the presence of stochastic gradient noise requires a smaller step size and, thus, induces higher gradient complexity to ensure convergence to an SOSP, compared to exact gradient descent (GD). In contrast, Liu et al. (2024) adopted a constant step size of $\frac{1}{M}$, which is only appropriate for GD with exact gradients and fails to account for the stochastic nature of population risk minimization. This misstep leads to an incorrect error rate. Specifically, the increased gradient complexity reduces the number of data points per gradient estimate, leading to larger estimation errors. Consequently, the correct error rate for achieving an SOSP should be looser than the one presented in Liu et al. (2024).

With the perception of the above gaps, we further argue that directly fixing the error in Liu et al. (2024) through a revised proof for their algorithm, while feasible, would fail to achieve the target SOSP with the optimal dependence on $\alpha$ as required in Definition 1. For a detailed discussion, please refer to Appendix B.

**Limitations in Private SOSP Selection**  The state-of-the-art learning algorithm proposed in Liu et al. (2024) only guarantees the existence of an $\alpha$-SOSP in all models throughout the learning process. To privately select an $\alpha$-SOSP from these iterates, the authors employ the well-known AboveThreshold algorithm. This approach raises significant concerns, as it relies on evaluating both the gradient and the Hessian matrix of the objective function for every model. In the case of population risk minimization, where the objective function is unknown and only samples from the data distribution are available, approximating these gradients and Hessians requires additional data and becomes computationally expensive—particularly for Hessians. Due to the need for computing the Hessian, this method is no longer first-order. Moreover, while Liu et al. (2024) claim that the error introduced by approximating the gradients and Hessians does not exceed the error generated by their learning algorithm, as we will demonstrate in Section 6, this assertion does not hold in distributed learning scenarios. The AboveThreshold algorithm primarily uses the perturbed gradient norm and the minimum eigenvalue of the Hessian for model selection. In single-machine cases, where all data is centrally stored, it is feasible to approximate the gradients and Hessians and add noise only to the one-dimensional values of the gradient norm and minimum eigenvalue. However, in distributed learning with heterogeneous data, each client must perturb and share its local gradients and Hessians—rather than just the one-dimensional quantities—so that noisy estimates can be aggregated at the central server. This introduces significantly more noise, especially in high-dimensional settings, thereby worsening the error rates provided by the learning algorithm (see Section 6 for a detailed analysis).

## 4 REVISIT PERTURBED SGD WITH GAUSSIAN NOISES

We begin by revisiting perturbed stochastic gradient descent (PSGD) with Gaussian noise for escaping saddle points in population risk minimization $F(\cdot)$. In PSGD, the model is updated iteratively. At each iteration $t$, instead of using the standard stochastic gradient oracle $g_t := \nabla F(x_{t-1}) + \hat{\zeta}_t$, where $\hat{\zeta}_t$ is the unbiased noise introduced by the stochastic gradient oracle, we introduce additional Gaussian noise to obtain a perturbed stochastic gradient oracle $\hat{g}_t$. The model update rule is:

$$x_t \leftarrow x_{t-1} - \eta \hat{g}_t, \tag{2}$$

where the perturbed stochastic gradient oracle $\hat{g}$ is defined as

$$\hat{g}_t := g_t + \hat{\xi}_t = \nabla F(x_{t-1}) + \hat{\zeta}_t + \hat{\xi}_t. \tag{3}$$

Here, the stochasticity in $\hat{g}_t$ arises from two sources: (i) $\hat{\zeta}_t$, the stochastic noise from the original stochastic gradient oracle, which typically depends on the (unknown) data distribution, and (ii) $\hat{\xi}_t$, a Gaussian noise term, $\hat{\xi}_t \sim \mathcal{N}(0, r^2 \mathbf{I}_d)$, added intentionally to facilitate escape from saddle points. Following prior works on stochastic optimization (Jin et al., 2021; Liu et al., 2024), we assume that $\hat{\zeta}_t \sim \mathrm{nSG}(\sigma)$, where nSG denotes the norm-sub-Gaussian distribution defined in Definition 6 in Appendix. Define $\psi := \sqrt{\sigma^2 + r^2 d}$, which captures the overall magnitude of noise in $\hat{g}_t$.

Note that we consider a different problem setting from prior work on PSGD Jin et al. (2021). In their setting, a target error $\alpha$ is specified, and the noise magnitude is adjusted accordingly to escape saddle points. However, in our case, privacy is the primary concern, and the Gaussian noise magnitude is determined by the DP budget. Therefore, our goal is to determine the error $\alpha$ achievable under a given privacy budget $(\epsilon, \delta)$ which fixes the Gaussian noise magnitude. The parameter settings and results from Jin et al. (2021) are not directly applicable to our setting. To guarantee a specific error $\alpha$, their method sets the Gaussian noise magnitude for perturbation such that $r^2 d = O(\sigma^2 + \alpha^{\frac{3}{2}})$, which is only valid in our setting when the noise magnitude $r$, as determined by the privacy budget, is sufficiently large, i.e., $r \geq O(\frac{\sigma}{\sqrt{d}})$. The behavior when $r$ is small remains an open question.

### 4.1 OUR APPROACH: A GENERAL GAUSSIAN-PERTURBED SGD FRAMEWORK

We introduce our framework in Algorithm 1. In this algorithm, we use a general stochastic gradient oracle with Gaussian perturbation, as described in (3), which we denote as `P_Grad_Oracle(*)` in steps 4 and 10, where $*$ omits any specific arguments the oracle might require. This allows our algorithm to serve as a general optimization framework for non-convex stochastic optimization, applicable beyond the context of DP. Building upon the PSGD updates described earlier, our algorithm distinguishes itself from the PSGD algorithm of Jin et al. (2021) by using the moving distance of the model parameters as the criterion for escaping saddle points (step 12). This innovation allows the algorithm to determine convergence to an SOSP with high probability during the PSGD process. In contrast, the algorithm proposed by Jin et al. (2021) outputs all model parameters obtained throughout the PSGD iterations, only guaranteeing that an SOSP was visited at least once. To further ensure the output is an SOSP with high probability, their method requires additional post-processing steps, such as computing the minimum eigenvalue of each Hessian matrix of the empirical risk function or approximating these eigenvalues using extra data samples for population risk. These steps introduce significant computational costs and extra sample usage, as well as the need to compute second-order information, thereby making the overall procedure no longer first-order.

We observe that, when successfully escaping from a saddle point, not only does the function value decrease sufficiently, as noted in Jin et al. (2021), but the model parameter also moves sufficiently far beyond a certain threshold $\mathscr{S}$ (specified later). Leveraging this key insight, our algorithm can directly identify and output an SOSP during the PSGD process, eliminating the need for additional calculations or sample usage.

### 4.2 ERROR RATE ANALYSIS FOR ALGORITHM 1

We begin by introducing the following algorithmic parameter setup and useful notations:

---

**Algorithm 1:** Gaussian Perturbed Stochastic Gradient Descent

**Input:** Failure probability $\omega$, initial model $x_0$, learning rate $\eta$, repeat number of the saddle point escape process $Q$, model deviation threshold $\mathscr{S}$, number of escape steps $\mathscr{T}$

1  $t \leftarrow 0$;
2  **while** true **do**
3      $t \leftarrow t + 1$;
4      $\hat{g}_t \leftarrow$ P_Grad_Oracle($*$);
5      **if** $\|\hat{g}_t\| \leq 3\chi$ **then**
        /* Saddle point escape                        */
6          $\tilde{t} \leftarrow t, \tilde{x} \leftarrow x_{t-1}, \mathsf{esc} \leftarrow$ false;
7          **for** $q \leftarrow 1, \cdots, Q$ **do**
8              $t \leftarrow \tilde{t}, x_t \leftarrow \tilde{x}$;
9              **for** $\tau \leftarrow 1, \cdots, \mathscr{T}$ **do**
10                 $\hat{g}_t \leftarrow$ P_Grad_Oracle($*$);
11                 $x_t \leftarrow x_{t-1} - \eta \cdot \hat{g}_t$;
12                 **if** $\|x_t - \tilde{x}\| \geq \mathscr{S}$ **then**
13                     $\mathsf{esc} \leftarrow$ true;
14                     **break**;
15                 **else**
16                     $t \leftarrow t + 1$;
17             **if** $\mathsf{esc}$ = true **then**
18                 **break**;
19         **if** $\mathsf{esc}$ = false **then**
20             **return** $x_{t-1}$
21     **else**
        /* Normal descent step                        */
22         $x_t \leftarrow x_{t-1} - \eta \cdot \hat{g}_t$;

---

$$\iota := s \cdot \mu, \qquad \chi := \max\left\{ 4\sqrt{C}s\mu^2, C\sqrt{2\log\frac{4T}{\omega}} \right\} \cdot \psi = 4\sqrt{C}s\mu^2\psi, \qquad \alpha := 4\chi,$$

$$\eta = \frac{\sqrt{\rho\alpha}}{M^2\iota^2} \leq \frac{1}{M}, \qquad \mathscr{T} := \frac{\iota}{s\eta\sqrt{\rho\alpha}}, \qquad \mathscr{S} := \frac{1}{\iota^{1.5}}\sqrt{\frac{\alpha}{\rho}}, \qquad \mathscr{F} := \frac{s}{8\iota^3}\sqrt{\frac{\alpha^3}{\rho}}, \tag{4}$$

where $s$ is a sufficiently large absolute constant to be determined later, and $\mu$ is a logarithmic factor defined as:

$$\mu = \max\left\{ \frac{1}{s}\log\left( \frac{9d}{C^{\frac{1}{4}}\eta\sqrt{s\rho\psi}}\log\left( \frac{4C^{\frac{1}{4}}}{s\eta r}\sqrt{\frac{\psi}{\rho}} \right) \right), \log\left( \frac{160\sqrt{2}C^{\frac{1}{4}}}{s\sqrt{\eta}r}\sqrt{\frac{\psi}{\rho}} \right), \frac{(C \cdot \log\frac{4T}{\omega})^{\frac{1}{4}}}{2^{\frac{3}{4}}\sqrt{s}}, 1 \right\}. \tag{5}$$

Throughout our analysis, $C$ represents an absolute constant that does not depend on $s$, and its value may change from line to line.

Let $\mathcal{H} := \nabla^2 F(\tilde{x})$, $v_{\min}$ be the eigenvector corresponding to the minimum eigenvalue of $\mathcal{H}$, and $\gamma := -\lambda_{\min}(\mathcal{H})$. Let $\mathcal{P}_{-v_{\min}}$ denote the projection onto the subspace orthogonal to $v_{\min}$.

**Definition 4** (Coupling Sequence). Let $\{x_i\}$ and $\{x_i'\}$ be two sequences obtained by separate runs of PSGD both starting at $\tilde{x}$. We say they are coupled if they share the same randomness for $\mathcal{P}_{-v_{\min}}\hat{\xi}_t$ and $\mathcal{B}_t$ at each iteration $t$, while in $v_{\min}$ direction, the random noise is opposite: $v_{\min}^\top\hat{\xi}_t = -v_{\min}^\top\hat{\xi}_t'$.

Our key insight that, when starting in the vicinity of any strict saddle point, PSGD will cause the model to drift sufficiently far away with high probability. See proof of Lemma 1 in Appendix D.1.

**Lemma 1** (Escaping Saddle Points). Consider two coupled sequence $\{x_i\}$ and $\{x_i'\}$, if $\tilde{x}$ satisfies $\|\nabla F(\tilde{x})\|_2 \leq \alpha$ and $\lambda_{\min}(\nabla^2 F(\tilde{x})) \leq -\sqrt{\rho\alpha}$, then with probability at least $\frac{1}{2}$, $\exists i \leq \mathscr{T}$ such that $\max\{\|x_i - \tilde{x}\|^2, \|x_i' - \tilde{x}\|^2\} \geq \mathscr{S}^2$.

**Corollary 1.** PSGD can successfully escape from any saddle point, moving the model at least a distance of $\mathscr{S}$ away from the saddle point, with a constant probability of at least $\frac{1}{4}$.

Corollary 1 guarantees only a constant probability of successful escape by PSGD. To boost this probability to meet any desired failure probability $\omega_0$, we can repeat the escape procedure independently up to $Q$ times, which corresponds to steps 7-18 in Algorithm 1.

**Lemma 2.** Given any target failure probability $\omega_0 \in (0,1)$, by repeating the $\mathscr{T}$-step PSGD process independently $Q = \frac{5}{2} \log \frac{1}{\omega_0}$ times, we can ensure successful escape with probability at least $1 - \omega_0$.

See proof of Lemma 2 in Appendix D.2. In the remainder of this section, we analyze the total number of PSGD steps required, in the worst case, for Algorithm 1 to reach a second-order stationary point. This is primarily determined by the decrease in the function value at each step.

We begin by presenting a standard result that shows how the change in function value can be decomposed into the decrease due to gradient magnitudes and a possible increase due to randomness in both the stochastic gradients and the perturbations. Let $\nu_t := \hat{\zeta}_t + \hat{\xi}_t$ represent the total noise.

**Lemma 3** (Descent Lemma). For any time step $t_0$, we have

$$F(x_{t_0+t}) - F(x_{t_0}) \leq -\frac{\eta}{2} \sum_{i=0}^{t-1} \|\nabla F(x_{t_0+i})\|^2 + \frac{\eta}{2} \sum_{i=1}^{t} \|\nu_{t_0+i}\|^2 \tag{6}$$

**Corollary 2.** There exists an absolute constant $c$ such that, for any given $t_0$, with probability at least $1 - 2e^{-\iota}$, we have

$$F(x_{t_0+t}) - F(x_{t_0}) \leq -\frac{\eta}{2} \sum_{i=0}^{t-1} \|\nabla F(x_{t_0+i})\|^2 + c \cdot \eta\psi^2(t + \iota). \tag{7}$$

Lemma 3 (see proof in Appendix D.3) and Corollary 2 (see proof in Appendix D.4) imply that large gradients lead to a rapid decrease in the function value. Next, we show in Lemma 4 that PSGD updates can significantly decreases the function value with high probability when starting near any strict saddle point and escaping it successfully. Proof of Lemma 4 is in Appendix D.5.

**Lemma 4** (Function Value Decrease per Successful Escape). Suppose a successful escape occurs after $\tau$ steps of PSGD initiated at $x_{t_0}$ ($\tau \leq \mathscr{T}$). With probability at least $1 - 2e^{-\iota}$, the function value decreases by at least $\frac{s}{8\iota^3} \sqrt{\frac{\alpha^3}{\rho}}$.

Next, we derive the maximum number of PSGD steps required in Lemma 6 (proved in Appendix D.7), which relies on the gradient estimation error given in Lemma 5 (proved in Appendix D.6).

**Lemma 5** (Gradient Estimation Error). With probability at least $\frac{\omega}{2}$, $\|\nu_t\| \leq C\sqrt{2\log\frac{4T}{\omega}}\psi \leq \chi$.

**Lemma 6** (Maximum Number of Descent Steps). Given the failure probability $\omega$, Algorithm 1 returns an $\alpha$-second-order stationary point within at most $\tilde{O}\left(\frac{1}{\alpha^{2.5}}\right)$ steps of PSGD updates.

**Remark 1.** At first glance, Lemma 6 seems to show an improvement in gradient complexity compared to Jin et al. (2021), reducing it from $O\left(\frac{1}{\alpha^4}\right)$ to $O\left(\frac{1}{\alpha^{2.5}}\right)$ for PSGD. However, we argue that our result is not directly comparable due to differences in the problem setting. Jin et al. (2021) assumes the target error is given as an input parameter, which can be arbitrarily small, and the gradient variance $\sigma$ is often treated as a constant. In contrast, we consider a scenario where the perturbation noise $r$ and the stochastic variance $\sigma$ are given, meaning the error rate $\alpha$ is determined by these parameters and cannot be made arbitrarily small. As a result, the gradient complexity of our algorithm should actually depend on $\sigma$ and $r$. For simplicity, we use $\alpha$ to describe the gradient complexity in our context.

In summary, we have the follow guarantee for Algorithm 1.

**Theorem 1.** Under Assumption 1 and 2, for any given $\omega \in (0,1)$, if the parameters are set as in (4), then with probability at least $1 - \omega$, Algorithm 1 will output an $\alpha$-second-order stationary point with an error of $\alpha = 4\chi$, using a maximum of $\tilde{O}\left(\frac{1}{\alpha^{2.5}}\right)$ steps of PSGD updates.

---

**Algorithm 2:** Adaptive DP-SPIDER

---

**Input:** privacy budget $\epsilon$ and $\delta$, time horizon $T$, models $\{x_{t-1}\}_{t=1}^T$, parameter $\kappa$

1   $t \leftarrow 1$, drift $\leftarrow \kappa$;
2   **while** $t \leq T$ **do**
3      **if** *drift* $\geq \kappa$ **then**
4          Sample mini-batch $\mathcal{B}_t$ of size $b_1$ from $\mathcal{D}$;
5          Sample $\xi_t \sim \mathcal{N}\left(0, c_1 \frac{G^2 \log \frac{1}{\delta}}{b_1^2 \epsilon^2} \mathbf{I}_d\right)$;
6          $\hat{g}_t \leftarrow \mathcal{O}_1(x_{t-1}, \mathcal{B}_t) + \xi_t$;
7          drift $\leftarrow 0$;
8      **else**
9          Sample mini-batch $\mathcal{B}_t$ of size $b_2$ from $\mathcal{D}$;
10          Sample $\xi_t \sim \mathcal{N}\left(0, c_2 \frac{M^2 \log \frac{1}{\delta}}{b_2^2 \epsilon^2} \|x_{t-1} - x_{t-2}\|^2 \mathbf{I}_d\right)$;
11          $\hat{g}_t \leftarrow \hat{g}_{t-1} + \mathcal{O}_2(x_{t-1}, x_{t-2}, \mathcal{B}_t) + \xi_t$;
12      drift $\leftarrow$ drift $+ \eta^2 \|\hat{g}_t\|^2$;
13      $t \leftarrow t + 1$;

**Output:** $\hat{g}_1, \hat{g}_2, \cdots, \hat{g}_T$

---

## 5   ERROR RATE FOR SOSP IN DP STOCHASTIC OPTIMIZATION

We now derive error rates for DP stochastic optimization based on the general result from Theorem 1. The error rate is derived using the adaptive DP-SPIDER algorithm, shown in Algorithm 2, as the gradient oracle. This adaptive version refines the original SPIDER by adjusting gradient queries based on model drift. Unlike standard SPIDER, which queries $\mathcal{O}_1$ at fixed intervals, potentially allowing the gradient estimation error to grow over time, adaptive SPIDER tracks the total model drift, defined as $\text{drift}_t := \sum_{i=\tau(t)}^t \|x_i - x_{i-1}\|^2$, where $\tau(t)$ is the last iteration when $\mathcal{O}_1$ was used.

The intuition behind is that, for smooth functions, the error in $\mathcal{O}_2$, which estimates $\nabla F(x_{t-1}) - \nabla F(x_{t-2})$, is proportional to $\|x_{t-1} - x_{t-2}\|$. When the model drift is small, the gradient estimate remains accurate enough, and $\mathcal{O}_2$ can continue to be used, reducing variance in gradient estimation (steps 9-11). However, when the drift grows large, further use of $\mathcal{O}_2$ could introduce significant error, and thus $\mathcal{O}_1$ is queried to refresh the gradient estimate (steps 4-7). A threshold $\kappa$ is set to determine when the drift becomes excessive, ensuring the total error remains controlled (step 3).

Our adaptive SPIDER differs from the approach in Liu et al. (2024) in an important way. In addition to triggering $\mathcal{O}_1$ when the model drift exceeds a threshold, Liu et al. (2024) also needs to add additional Gaussian noise and trigger $\mathcal{O}_1$ every time a potential saddle point is reached, while our method fully utilizes the DP Gaussian noise already present in the gradient oracle, avoiding the need to add additional noise at saddle points. As a result, in our framework, the decision to query $\mathcal{O}_1$ or continue with $\mathcal{O}_2$ is based solely on model drift, leading to a simpler and more efficient gradient estimation process. The following Lemma 7 captures the noise magnitude for any $\hat{g}_t$, which is proved in Appendix E.1.

**Lemma 7.** Under Assumption 1, for all $t \in [T]$, our adaptive DP-SPIDER guarantees that $\hat{g}_t$ satisfies (3) with:

$$\sigma \leq O\left(\sqrt{\frac{G^2 \log^2 d}{b_1} + \frac{M^2 \log^2 d}{b_2} \kappa}\right), \qquad r \leq O\left(\sqrt{\frac{G^2 \log \frac{1}{\delta}}{b_1^2 \epsilon^2} + \frac{M^2 \log \frac{1}{\delta}}{b_2^2 \epsilon^2} \kappa}\right). \tag{8}$$

We can bound the number of occurrences where the drift becomes large, which allows us to limit the total number of queries to $\mathcal{O}_1$. This enables the proper setting of $b_1$ and $b_2$ for Algorithm 2.

**Lemma 8.** Under Assumption 1 and 2, let $\mathcal{T} := \{t \in [T] : \text{drift}_t \geq \kappa\}$ be the set of rounds where the drift exceeds the threshold $\kappa$. Under the same probability as in Theorem 1, $|\mathcal{T}| \leq O\left(\frac{U\eta}{\kappa}\right)$.

**Theorem 2.** Under Assumption 1 and 2, with $\sigma$ and $r$ set as determined by Lemma 7, let Algorithm 1 run with the gradient oracle instantiated by Algorithm 2, where $b_1 = \frac{n\kappa}{2U\eta}$, $b_2 = \frac{n\eta\chi^2}{2U}$ and $\kappa = $

---

**Algorithm 3:** Distributed Adaptive DP-SPIDER

**Input:** privacy budget $\epsilon$ and $\delta$, time horizon $T$, models $\{x_{t-1}\}_{t=1}^T$, parameter $\kappa$

1   $t \leftarrow 1$, drift $\leftarrow \kappa$;
2   **while** $t \leq T$ **do**
3     **if** *drift* $\geq \kappa$ **then**
4       **for every** *client* $j$ **in parallel do**
5         Sample mini-batch $\mathcal{B}_{j,t}$ of size $b_1$ from $\mathcal{D}_j$;
6         Sample $\xi_{j,t} \sim \mathcal{N}\left(0, c_1 \frac{G^2 \log\frac{1}{\delta}}{b_1^2 \epsilon^2} \mathbf{I}_d\right)$;
7         $\hat{g}_{j,t} \leftarrow \mathcal{O}_1(x_{t-1}, \mathcal{B}_{j,t}) + \xi_{j,t}$;
8         Send $\hat{g}_{j,t}$ to the server;
9       drift $\leftarrow 0$;
10    **else**
11       **for every** *client* $i$ **in parallel do**
12         Sample mini-batch $\mathcal{B}_{j,t}$ of size $b_2$ from $\mathcal{D}_j$;
13         Sample $\xi_{j,t} \sim \mathcal{N}\left(0, c_2 \frac{M^2 \log\frac{1}{\delta}}{b_2^2 \epsilon^2} \|x_{t-1} - x_{t-2}\|^2 \mathbf{I}_d\right)$;
14         $\hat{g}_{j,t} \leftarrow \hat{g}_{j,t-1} + \mathcal{O}_2(x_{t-1}, x_{t-2}, \mathcal{B}_{j,t}) + \xi_{j,t}$;
15         Send $\hat{g}_{j,t}$ to the server;
16    $\hat{g}_t \leftarrow \frac{1}{m} \sum_{j=1}^m \hat{g}_{j,t}$;
17    drift $\leftarrow$ drift $+ \eta^2 \|\hat{g}_t\|^2$;
18    $t \leftarrow t + 1$;

**Output:** $\hat{g}_1, \hat{g}_2, \cdots, \hat{g}_T$

---

$\max\left\{ \frac{G^{\frac{3}{2}} U^{\frac{1}{2}} \rho^{\frac{1}{2}}}{M^{\frac{5}{2}} n^{\frac{1}{2}}}, \frac{G^{\frac{14}{15}} d^{\frac{2}{5}} U^{\frac{4}{5}} \rho^{\frac{8}{15}}}{M^{\frac{34}{15}} (n\epsilon)^{\frac{4}{5}}} \right\}$. Then the whole algorithm satisfies $(\epsilon, \delta)$-DP with some constants

$c_1, c_2$ and finds an $\alpha$-SOSP with $\alpha = \tilde{O}\left( \frac{1}{n^{\frac{1}{3}}} + \left( \frac{\sqrt{d}}{n\epsilon} \right)^{\frac{2}{5}} \right)$.

See proof of Lemma 8 in Appendix E.2, and proof of Theorem 2 in Appendix E.3.

## 6   APPLICATION TO DISTRIBUTED SGD

Our framework, Algorithm 1, can be extended to distributed learning scenarios by adapting the gradient oracles from Algorithm 2 to a distributed version, as presented in Algorithm 3. Our algorithm can be viewed as an adaptive improvement of the previous DIFF2 algorithm Murata & Suzuki (2023) for distributed learning, which uses standard SPIDER to converge to first-order stationary points and is limited to handling homogeneous data. To the best of our knowledge, our algorithm is the first to achieve differentially private distributed learning that not only guarantees convergence to second-order stationary points but also operates effectively in heterogeneous data settings. The following Lemma 9 captures the noise magnitude for any $\hat{g}_t$ given by Algorithm 3, which is proved in Appendix F.1.

**Lemma 9.** Under Assumption 1, for $\forall t \in [T]$, our distributed adaptive DP-SPIDER guarantees that $\hat{g}_t$ satisfies (3) with

$$\sigma \leq O\left( \sqrt{\frac{G^2 \log^2 d}{m \cdot b_1} + \frac{M^2 \log^2 d}{m \cdot b_2} \kappa} \right), \qquad r \leq O\left( \sqrt{\frac{G^2 \log\frac{1}{\delta}}{m \cdot b_1^2 \epsilon^2} + \frac{M^2 \log\frac{1}{\delta}}{m \cdot b_2^2 \epsilon^2} \kappa} \right). \quad (9)$$

**Theorem 3.** Under Assumption 1 and 2, with $\sigma$ and $r$ set as determined by Lemma 9, let Algorithm 1 run with the gradient oracle instantiated by Algorithm 3, where $b_1 = \frac{n\kappa}{2U\eta}$, $b_2 = \frac{n\eta\chi^2}{2U}$ and $\kappa = \max\left\{ \frac{G^{\frac{3}{2}} U^{\frac{1}{2}} \rho^{\frac{1}{2}}}{M^{\frac{5}{2}} (mn)^{\frac{1}{2}}}, \frac{G^{\frac{14}{15}} d^{\frac{2}{5}} U^{\frac{4}{5}} \rho^{\frac{8}{15}}}{M^{\frac{34}{15}} (\sqrt{m}n\epsilon)^{\frac{4}{5}}} \right\}$. Then the whole algorithm satisfies $(\epsilon, \delta)$-ICRL-DP with some constants $c_1, c_2$, and finds an $\alpha$-SOSP with $\alpha = \tilde{O}\left( \frac{1}{(mn)^{\frac{1}{3}}} + \left( \frac{\sqrt{d}}{\sqrt{m}n\epsilon} \right)^{\frac{2}{5}} \right)$.

---

**Algorithm 4:** Distributed Private Model Selection

**Input:** A group of models $\{x_t\}_{t=1}^T$, privacy parameters $\epsilon, \delta$

1 **for** $t \leftarrow 1, \cdots, T$ **do**

2    **for** every *client* $j$ **in parallel do**

3       Compute $\nabla \bar{F}_j(x_t) \leftarrow \nabla f(x_t; S_j) + \theta_{i,t}$, where $\theta_{i,t} \sim \mathcal{N}\left(0, c_1 \frac{G^2 T \log \frac{1}{\delta}}{n^2 \epsilon^2} \mathbf{I}_d\right)$ ;

4       Compute $\nabla^2 \bar{F}_j(x_t) \leftarrow \nabla^2 f(x_t; S_j) + \mathbf{H}_{j,t}$, where $\mathbf{H}_{j,t}$ is a symmetric matrix with its upper triangle (including the diagonal) being i.i.d. samples from $\mathcal{N}\left(0, c_2 \frac{M^2 dT \log \frac{1}{\delta}}{n^2 \epsilon^2}\right)$ and each lower triangle entry is copied from its upper triangle counterpart;

5       Send $\nabla \bar{F}_j(x_t)$ and $\nabla^2 \bar{F}_j(x_t)$ to the server;

6    $\nabla \bar{F}(x_t) \leftarrow \frac{1}{m} \sum_{j=1}^m \nabla \bar{F}_j(x_t), \nabla^2 \bar{F}(x_t) \leftarrow \frac{1}{m} \sum_{j=1}^m \nabla^2 \bar{F}_j(x_t);$

7    **if** $\|\nabla \bar{F}(x_t)\|_2 \leq \alpha + \frac{G \log\left(\frac{8d}{\omega'}\right)}{\sqrt{mn}} + \frac{G\sqrt{dT \log\left(\frac{1}{\delta}\right) \log\left(\frac{16}{\omega'}\right)}}{\sqrt{mn}\epsilon}$ **and**

     $\lambda_{\min}\left(\nabla^2 \bar{F}(x_t)\right) \geq -\left(\sqrt{\rho\alpha} + M\sqrt{\frac{\log\left(\frac{8d}{\omega'}\right)}{mn}} + \frac{Md\sqrt{T \log \frac{1}{\delta} \log\left(\frac{32}{\omega'}\right)}}{\sqrt{mn}\epsilon}\right)$ **then**

8       **Return** $x_t$

---

The proof of Lemma 9 is in Appendix F.1 and the proof of Theorem 3 is in Appendix F.2.

**Remark 2.** Our convergence error rate reflects a collaborative synergy between clients, indicating that our algorithm significantly benefits from the distributed framework. Specifically, there is a linear term in $m$ before $n$ in the first non-private term of $\alpha$, and a square root term $\sqrt{m}$ before $n$ in the second term, which accounts for the privacy cost. This separation arises due to data heterogeneity, and the synergy effect aligns with that observed in other DP distributed learning works under heterogeneity, such as Gao et al. (2024).

Finally, we discuss in detail the superiority of our proposed algorithm framework for distributed learning scenarios. If we cannot guarantee the output of an SOSP with high probability, as we do in Algorithm 1, then, like the approach given in Liu et al. (2024), we would need to rely on some private model selection algorithm to evaluate all the models obtained during the iterations and select an SOSP. The state-of-the-art private selection method is the AboveThreshold algorithm, as used by Liu et al. (2024). We extend this algorithm to the distributed setting, as presented in Algorithm 4. Now, suppose we have a list of model iterates $x_{t t \in T}$ output by a learning algorithm, with at least one point being an $\alpha$-SOSP. We then provide the following error rate guarantee for the model selected by Algorithm 4, whose proof in given in Appendix F.3.

**Theorem 4.** Algorithm 4 is $(\epsilon, \delta)$-ICRL-DP. Under Assumption 1, if $mn \geq \frac{4}{9} \log \frac{8d}{\omega'}$, then with probability at least $1 - \omega'$, we have the following two holds for Algorithm 4.

- If there exists an $\alpha$-SOSP point $x_p \in \{x_t\}_{t=1}^T$, then Algorithm 4 will output one point.

- If Algorithm 4 outputs any point $x_o$, then $x_o$ is an $\alpha'$-SOSP with

$$\alpha' = \tilde{O}\left(\alpha + \frac{1}{mn} + \frac{1}{\sqrt{mn}} + \frac{\alpha}{\sqrt{mn}} + \frac{\sqrt{d}}{\sqrt{mn}\epsilon\alpha^{\frac{5}{4}}} + \frac{d}{\sqrt{mn}\epsilon\alpha^{\frac{3}{4}}} + \frac{d^2}{mn^2\epsilon^2\alpha^{\frac{5}{2}}}\right). \quad (10)$$

**Remark 3.** To ensure that Algorithm 4 outputs a model with error no worse than that guaranteed by the learning algorithm, i.e., $\alpha$, it is necessary to satisfy the condition $\tilde{O}\left(\frac{\sqrt{d}}{\sqrt{mn}\epsilon\alpha^{\frac{5}{4}}} + \frac{d}{\sqrt{mn}\epsilon\alpha^{\frac{3}{4}}} + \frac{d^2}{mn^2\epsilon^2\alpha^{\frac{5}{2}}}\right) \leq \tilde{O}(\alpha)$, since $\tilde{O}\left(\frac{1}{mn} + \frac{1}{\sqrt{mn}} + \frac{\alpha}{\sqrt{mn}}\right) \leq \tilde{O}(\alpha)$ holds trivially. This reduces to a constraint on the model dimension $d$, such that $d \leq \min\{(\sqrt{mn}\epsilon)^2, (\sqrt{mn}\epsilon)^{\frac{6}{13}}\}$. Therefore, it is impractical to apply private model selection in distributed learning scenarios, especially when the model dimension is large.

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

## A   RELATED WORK

**Private Stochastic Optimization**   Differential privacy (DP) has become a crucial component in stochastic optimization due to increasing concerns about data privacy. The pioneering work by Dwork et al. (2006) established the foundational principles of DP, and its application in stochastic optimization has since seen significant progress. Early efforts primarily focused on convex optimization, achieving strong privacy guarantees while ensuring efficient learning, with a long list of representative works e.g., (Wang et al., 2020a; 2017; Bassily et al., 2014; 2019; 2021; Feldman et al., 2020; Choquette-Choo et al., 2024; Wang et al., 2018; Su et al., 2023; 2024; Hu et al., 2022; Xue et al., 2021; Wang et al., 2020b; Huai et al., 2020). Recent advances have extended DP to non-convex settings, mainly focusing on first-order stationary points (FOSP). Notable works in this area include (Arora et al., 2023; Bassily et al., 2021; Zhou et al., 2020; Wang et al., 2019; Xiao et al., 2023), which improved error rates in non-convex optimization with balanced privacy and utility in stochastic gradient methods. However, these works generally fail to address the more stringent criterion of second-order stationary points (SOSP). The very recent work Liu et al. (2024) tired to narrow this gap, but unfortunately has some issues in their results as we discussed before. Our work builds on this foundation by correcting error rates and proposing a framework that ensures convergence to SOSP while maintaining DP.

**Finding Second-Order Stationary Points (SOSP)**   In non-convex optimization, convergence to FOSP is often insufficient, as saddle points can lead to sub-optimal solutions (Jain et al., 2015; Sun et al., 2016). Achieving SOSP, where the gradient is small and the Hessian is positive semi-definite, ensures that the optimization converges to a local minimum rather than a saddle point. Techniques for escaping saddle points, such as perturbed SGD with Gaussian noise, have been explored in works like Jin et al. (2021) and Ge et al. (2015). Ge et al. (2015) first showed that SGD with a simple parameter perturbation can escape saddle points efficiently. Later, the analysis was refined by Jin et al. (2017; 2021). Recently, variance reduction techniques have been applied to second-order guaranteed methods Ge et al. (2019); Li (2019).These methods ensure escape from saddle points by introducing noise to the gradient descent process. In contrast, the studies of SOSP under DP are quite limited, and most of them only consider the empirical risk minimization objective, such as Wang et al. (2019); Wang & Xu (2021). Very recently, Liu et al. (2024) addressed the population risk minimization objective, but with notable gaps in their error analysis, particularly in the treatment of gradient variance. Moreover, all of these works are limited to the centralized learning setting with only one client and cannot be directly extended to the more general distributed learning setting.

**Distributed Learning**   Distributed learning has gained prominence due to the growing need for large-scale models trained on decentralized data. Methods like federated learning (McMahan et al., 2017) have enabled multiple clients to collaboratively train models without sharing their local data, preserving privacy. Recent efforts, such as Gao et al. (2024); Lowy et al. (2023); Lowy & Razaviyayn (2023) investigated DP learning problems in distributed settings, but these works are limited to first-order optimization. No prior work, to our knowledge, has extended these methods to ensure SOSP in distributed learning scenarios with heterogeneous data. Our proposed framework addresses this gap by introducing the first distributed learning algorithm with DP guarantees for SOSP, capable of handling arbitrary data heterogeneity across clients.

## B   MORE DISCUSSIONS ON GAPS IN SOTA

In this section, we further discuss whether the error in (Liu et al., 2024) can be directly fixed through a revised proof for their algorithm. While this is feasible, such a correction would still fail to achieve the target SOSP with the optimal dependence on $\alpha$ required in our work: $\|\nabla F(x)\| \leq \alpha$ and $\nabla^2 F(x) \succeq -\sqrt{\rho\alpha} \cdot \mathbf{I}_d$. Specifically, a direct correction would result in suboptimal second-order accuracy with a dependence of $\widetilde{O}(\alpha^{2/5})$, instead of the desired $\widetilde{O}(\alpha^{1/2})$.

The algorithm in (Liu et al., 2024) can be viewed as a special single-machine case of the generic framework of perturbed gradient descent (GD) with bounded gradient inexactness, as developed by Yin et al. (2019). In this view, DP noise contributes to bounded gradient inexactness. The analysis by Yin et al. (2019) implies the corrected convergence guarantees for the algorithm of Liu et al. (2024). Assuming the first-order error rate satisfies $\|\nabla F(x)\| \leq O(\alpha)$, the analysis in (Yin et al.,

2019) guarantees $\|\nabla F(x)\| \leq O(\alpha)$ and $\nabla^2 F(x) \succeq -\widetilde{O}(\sqrt{\rho}\alpha^{2/5}) \cdot \mathbf{I}_d$, see Theorem 3 of (Yin et al., 2019). However, this falls short of the desired guarantee of $\nabla^2 F(x) \succeq -\widetilde{O}(\sqrt{\rho\alpha}) \cdot \mathbf{I}_d$, which is also what Liu et al. (2024) ideally aimed to achieve.

Furthermore, Proposition 1 in (Yin et al., 2019) establishes a lower bound of $\widetilde{O}(\alpha^{1/2})$ for dependence on $\alpha$ in second-order guarantees, highlighting the suboptimality of $\widetilde{O}(\alpha^{2/5})$. Additionally, Theorem 4 in (Yin et al., 2019) shows that with an exact gradient oracle, an optimal dependence of $\widetilde{O}(\alpha^{1/2})$ can be achievable. This explains why Liu et al. (2024) appeared to achieve the optimal order, as their analysis omitted the effect of gradient variance, as we discussed in Section 3 of our paper.

In summary, while directly correcting the results of Liu et al. (2024) using a refined analysis is feasible and can be accomplished with minimal effort based on (Yin et al., 2019), such corrections still cannot guarantee the target SOSP. Designing a new framework, as we have done, is therefore both necessary and essential to meet these expectations.

## C  USEFUL FACTS

### C.1  PROBABILITY TOOLS

**Definition 5** (Sub-Gaussian random vector (Jin et al., 2019, Definition 2))**.** A random vector $v \in \mathbb{R}^d$ is $\zeta$-*sub-Gaussian* (or $\mathrm{SG}(\zeta)$), if there exists a positive constant $\zeta$ such that

$$\mathbb{E}[\exp(\langle u, v - \mathbb{E}[v]\rangle)] \leq \exp\left(\frac{\|u\|_2^2 \zeta^2}{2}\right), \qquad \forall u \in \mathbb{R}^d. \tag{11}$$

**Definition 6** (Norm-sub-Gaussian random vector (Jin et al., 2019, Definition 3))**.** A random vector $v \in \mathbb{R}^d$ is $\zeta$-*norm-sub-Gaussian* (or $\mathrm{nSG}(\zeta)$), if there exists a positive constant $\zeta$ such that

$$\mathbb{P}\left[\|v - \mathbb{E}[v]\| \geq t\right] \leq 2\exp\left(-\frac{t^2}{2\zeta^2}\right), \qquad \forall t \in \mathbb{R}. \tag{12}$$

Note that norm-sub-Gaussian random vectors (Definition 6) are more general than sub-Gaussian random vectors (Definition 5), as sub-Gaussian distributions require *isotropy*, whereas norm-sub-Gaussian distributions do not impose this condition.

**Lemma 10** ((Jin et al., 2019, Lemma 1))**.** A $\mathrm{SG}(r)$ random vector $v \in \mathbb{R}^d$ is also $\mathrm{nSG}(2\sqrt{2} \cdot r\sqrt{d})$.

We are interested in the properties of norm-subGaussian martingale difference sequences. Concretely, they are sequences satisfying the following properties.

**Condition 1.** Consider random vectors $v_1, \cdots, v_p \in \mathbb{R}^d$, and corresponding filtrations $\mathcal{F}_i = \sigma(v_1, \cdots, v_i)$ for $i \in [n]$, such that $v_i | \mathcal{F}_{i-1}$ is zero-mean $\mathrm{nSG}(\zeta_i)$ with $\zeta_i \in \mathcal{F}_{i-1}$. That is,

$$\mathbb{E}[v_i | \mathcal{F}_{i-1}] = 0, \qquad \mathbb{P}\left[\|v_i\| \geq t | \mathcal{F}_{i-1}\right] \leq 2\exp\left(-\frac{t^2}{2\zeta^2}\right), \qquad \forall t \in \mathbb{R}, \forall i \in [p]. \tag{13}$$

**Lemma 11** (Hoeffding type inequality for norm-sub-Gaussian (Jin et al., 2019, Corollary 7))**.** Let random vectors $v_1, \cdots, v_p \in \mathbb{R}^d$, and corresponding filtrations $\mathcal{F}_i = \sigma(v_1, \cdots, v_i)$ for $i \in [k]$ satisfy condition 1 with fixed $\{\zeta_i\}$. Then for any $\iota > 0$, there exists an absolute constant $C$ such that, with probability at least $1 - 2d \cdot e^{-\iota}$,

$$\left\|\sum_{i=1}^{p} v_i\right\|_2 \leq C \cdot \sqrt{\sum_{i=1}^{p} \zeta_i^2 \cdot \iota}. \tag{14}$$

Lemma 11 implies that the sum of norm-sub-Gaussian random vectors is till norm-sub-Gaussian.

**Corollary 3.** Let random vectors $v_1, \cdots, v_p \in \mathbb{R}^d$, and corresponding filtrations $\mathcal{F}_i = \sigma(v_1, \cdots, v_i)$ for $i \in [k]$ satisfy condition 1 with fixed $\{\zeta_i\}$. Then $\sum_{i=1}^{p} v_i$ is $\mathrm{nSG}\left(C \cdot \sqrt{\log(d) \sum_{i=1}^{k} \zeta_i^2}\right)$.

*Proof.* Let $\zeta_+ := \sqrt{C \log(d) \sum_{i=1}^k \zeta_i}$. According to Definition 6, we aim to show that, for any $\omega \in (0,1)$, with probability at least $1 - \omega$, $\|\sum_{i=1}^p v_i\| \leq \sqrt{2\zeta_+^2 \ln \frac{2}{\omega}}$. By Lemma 11, we have known that, with probability at least $1 - \omega$, $\|\sum_{i=1}^p v_i\| \leq C \cdot \sqrt{\sum_{i=1}^p \zeta_i^2 \ln \frac{2d}{\omega}}$. Next, we show that $\sqrt{2\zeta_+^2 \ln \frac{2}{\omega}} \geq C \cdot \sqrt{\sum_{i=1}^p \zeta_i^2 \ln \frac{2d}{\omega}}$, which, by re-arranging the terms, is equivalent to show $\zeta_+^2 \geq \frac{C^2}{2}(\sum_{i=i}^p \zeta_i^2) \frac{\log \frac{2d}{\omega}}{\log \frac{2}{\omega}}$. This follows directly from the fact that $\frac{\log \frac{2d}{\omega}}{\log \frac{2}{\omega}} \leq 2 \log d, \forall \omega \in (0,1)$. $\square$

**Lemma 12** ((Jin et al., 2021, Lemma C.6)). Let random vectors $v_1, \cdots, v_p \in \mathbb{R}^d$, and corresponding filtrations $\mathcal{F}_i = \sigma(v_1, \cdots, v_i)$ for $i \in [k]$ satisfy condition 1, then for any $\iota > 0$, and $B > b > 0$, there exists an absolute constant $C$ such that, with probability at least $1 - 2d \log\left(\frac{B}{b}\right) \cdot e^{-\iota}$,

$$\sum_{i=1}^p \zeta_i^2 \geq B \qquad \text{or} \qquad \left\|\sum_{i=i}^p v_i\right\| \leq C \cdot \sqrt{\max\left\{\sum_i^p \zeta_i^2, b\right\} \cdot \iota}. \tag{15}$$

**Lemma 13** ((Jin et al., 2021, Lemma C.7)). Let random vectors $v_1, \cdots, v_p \in \mathbb{R}^d$, and corresponding filtrations $\mathcal{F}_i = \sigma(v_1, \cdots, v_i)$ for $i \in [k]$ satisfy condition 1 with fixed $\zeta_1 = \zeta_2 = \cdots = \zeta_p = \zeta$, then there exists an absolute constant $C$ such that, for any $\iota > 0$, with probability at least $1 - e^{-\iota}$,

$$\sum_{i=1}^p \|v_i\|^2 \leq C \cdot \zeta^2 \cdot (p + \iota). \tag{16}$$

**Lemma 14** (Matrix Bernstein inequality (Tropp, 2012, Theorem 1.4)). Consider a finite sequence $\{\mathbf{M}_i\}_{i \in [k]}$ of independent, random, self-adjoint matrices with dimension $d \times d$. Assume that each random matrix satisfies $\mathbb{E}[\mathbf{M}_i] = \mathbf{0}$, $\|\mathbf{M}_i\|_2 \leq B$, then for all $t \geq 0$, we have

$$\mathbb{P}\left[\left\|\sum_{i \in [k]} \mathbf{M}_i\right\|_2 \geq t\right] \leq d \exp\left(-\frac{t^2}{2(\sigma^2 + Bt/3)}\right), \tag{17}$$

where $\sigma^2 = \left\|\sum_{i \in [k]} \mathbb{E}[\mathbf{M}_i^2]\right\|_2$.

**Lemma 15** (Norm of symmetric matrices with sub-gaussian entries (Vershynin, 2020, Corollary 4.4.8)). Let $\mathbf{M}$ be an $d \times d$ symmetric random matrix whose entries $\mathbf{M}_{i,j}$ on and above the diagonal are independent, mean zero, sub-gaussian random variables. Then, with probability at least $1 - 4 \exp(-t^2)$, for any $t > 0$ we have

$$\|\mathbf{M}\|_2 \leq C \cdot \max_{i,j} \|\mathbf{M}_{i,j}\|_{\psi_2} \cdot (\sqrt{d} + t), \tag{18}$$

where $C$ is a universal constant.

## C.2 PRIVACY PRELIMINARIES

**Definition 7** (Gaussian Mechanism Dwork et al. (2014)). Given any input data $D \in \mathcal{X}^n$ and a query function $q : \mathcal{X}^n \to \mathbb{R}^d$, the Gaussian mechanism $\mathcal{M}_G$ is defined as $q(D) + \nu$ where $\nu \sim \mathcal{N}(0, \sigma_G^2 \mathbf{I}_d)$. Let $\Delta_2(q)$ be the $\ell_2$-sensitivity of $q$, *i.e.*, $\Delta_2(q) := \sup_{D \sim D'} \|q(D) - q(D')\|_2$. For any $\sigma, \delta > 0$, $\mathcal{M}_G$ guarantees $(\frac{\Delta_2(q)}{\sigma_G}\sqrt{2 \log \frac{1.25}{\delta}}, \delta)$-DP. That is, if we want the output of $q$ to be $(\epsilon, \delta)$-DP for any $0 < \epsilon, \delta < 1$, then $\sigma_G$ should be set to $\frac{\Delta_2(q)}{\epsilon}\sqrt{2 \log \frac{1.25}{\delta}}$.

**Lemma 16** (Parallel Composition of DP McSherry (2009)). Suppose there are $n$ $(\epsilon, \delta)$-differentially private mechanisms $\{\mathcal{M}_i\}_{i=1}^n$ and $n$ disjoint datasets denoted by $\{D_i\}_{i=1}^n$. Then the algorithm, which applies each $\mathcal{M}_i$ on the corresponding $D_i$, preserves $(\epsilon, \delta)$-DP in total.

# D OMITTED PROOFS IN SECTION 4

## D.1 PROOF OF LEMMA 1

*Proof of Lemma 1.* We define the following notations for proving Lemma 1.

$$\hat{x}_t := x_t - x'_t, \tag{19}$$

$$\hat{\zeta}_t := \hat{\zeta}_t - \hat{\zeta}'_t, \tag{20}$$

$$\hat{\xi}_t := \hat{\xi}_t - \hat{\xi}'_t, \tag{21}$$

$$\Delta_t := \int_0^1 \nabla^2 F(y \cdot x_t + (1-y) \cdot x'_t) \ \mathrm{d}y - \mathcal{H} \tag{22}$$

**Lemma 17** (Dynamics of the Coupling Sequence Difference). For any time step $t \geq 0$, we have

$$\hat{x}_t = -\eta \underbrace{\sum_{i=1}^t (\mathbf{I}_d - \eta\mathcal{H})^{t-i} \Delta_{i-1} \hat{x}_{i-1}}_{\mathscr{P}_h(t)} - \eta \underbrace{\sum_{i=1}^t (\mathbf{I}_d - \eta\mathcal{H})^{t-i} \hat{\zeta}_i}_{\mathscr{P}_{sg}(t)} - \eta \underbrace{\sum_{i=1}^t (\mathbf{I}_d - \eta\mathcal{H})^{t-i} \hat{\xi}_i}_{\mathscr{P}_p(t)}. \tag{23}$$

*Proof of Lemma 17.*

$$\hat{x}_t = x_t - x'_t = \hat{x}_{t-1} - \eta[\nabla F(x_{t-1}) - \nabla F(x'_{t-1}) + \hat{\zeta}_t - \hat{\zeta}'_t + \hat{\xi}_t - \hat{\xi}'_t] \tag{24}$$

$$= \hat{x}_{t-1} - \eta[(\mathcal{H} + \Delta_{t-1})\hat{x}_{t-1} + \hat{\zeta}_t + \hat{\xi}_t] = (\mathbf{I}_d - \eta\mathcal{H})\hat{x}_{t-1} - \eta[\Delta_{t-1}\hat{x}_{t-1} + \hat{\zeta}_t + \hat{\xi}_t] \tag{25}$$

$$= (\mathbf{I}_d - \eta\mathcal{H})^t \hat{x}_0 - \eta \sum_{i=1}^t (\mathbf{I}_d - \eta\mathcal{H})^{t-i}(\Delta_{i-1}\hat{x}_{i-1} + \hat{\zeta}_i + \hat{\xi}_i) \tag{26}$$

$$= -\eta \sum_{i=1}^t (\mathbf{I}_d - \eta\mathcal{H})^{t-i}(\Delta_{i-1}\hat{x}_{i-1} + \hat{\zeta}_i + \hat{\xi}_i), \tag{27}$$

where the last equality is due to $\hat{x}_0 = 0$. $\qquad\square$

We prove Lemma 1 by contradiction. Suppose that for $\forall t \leq \mathcal{T}$:

$$\max\left\{\|x_t - \tilde{x}\|^2, \|x'_t - \tilde{x}\|^2\right\} \leq \mathcal{S}^2. \tag{28}$$

With the above assumption (28), we show that $\mathscr{P}_p(t)$ controls the behavior of the dynamics, while $\mathscr{P}_h(t)$ and $\mathscr{P}_{sg}(t)$ remain small compared with $\mathscr{P}_p(t)$.

Define $\alpha := \sqrt{\sum_{i=1}^t (1 + \eta\gamma)^{2(t-i)}}$ and $\beta := (1 + \eta\gamma)^t / \sqrt{2\eta\gamma}$. It is easy to verify that $\alpha(t) \leq \beta(t)$ for any $t \in \mathbb{N}$.

**Lemma 18.** For $\forall t \geq 0$, we have

$$\mathbb{P}\left[\|\mathscr{P}_p(t)\| \leq c\beta(t)\eta r \cdot \sqrt{\iota}\right] \geq 1 - 2e^{-\iota} \tag{29}$$

$$\mathbb{P}\left[\|\mathscr{P}_p(t)\| \geq \frac{\beta(\mathcal{T})\eta r}{10}\right] \geq \frac{2}{3} \tag{30}$$

**Lemma 19.** If for $\forall t \leq \mathcal{T}$, $\max\left\{\|x_t - \tilde{x}\|^2, \|x'_t - \tilde{x}\|^2\right\} \leq \mathcal{S}^2$ holds, then we have

$$\mathbb{P}\left[\|\mathscr{P}_h(t) + \mathscr{P}_{sg}(t)\| \leq \frac{\beta(t)\eta r}{20}\right] \geq 1 - 6d\mathcal{T} \log\left(\frac{\mathcal{S}}{\eta r}\right) e^{-\iota} \tag{31}$$

*Proof of Lemma 19.* Denote by $\mathcal{E}$ the event $\{\forall t \leq \mathcal{T} : \max\left\{\|x_t - \tilde{x}\|^2, \|x'_t - \tilde{x}\|^2\right\} \leq \mathcal{S}^2\}$. We prove the following claim for any $t \leq \mathcal{T}$ by induction:

$$\mathbb{P}\left[\mathcal{E} \implies \forall i \leq t : \|\mathscr{P}_h(i) + \mathscr{P}_{sg}(i)\| \leq \frac{\beta(i)\eta r}{20}, \|\mathscr{P}_p(t)\| \leq c\beta(t)\eta r\sqrt{\iota}\right] \leq 1 - 6dt \log\left(\frac{\mathcal{S}}{\eta r}\right) e^{-\iota} \tag{32}$$

For the base case of $t = 0$, the claim holds trivially as $\mathscr{P}_h(0) = \mathscr{P}_{sg}(0) = 0$. Suppose there exists some $\tau < \mathscr{T}$ such that the claim holds for all $t \le \tau$, we then forward prove that the claim also holds for $t = \tau + 1 \le \mathscr{T}$. Since for any $t \le \tau$, $\|\mathscr{P}_p(t)\| \le c\beta(t)\eta r\sqrt{\iota}$, we have

$$\|\hat{x}_t\| \le \|\mathscr{P}_h(t) + \mathscr{P}_{sg}(t)\| + \|\mathscr{P}_p(t)\| \tag{33}$$

$$\le \frac{\beta(t)\eta r}{20} + c\beta(t)\eta r \cdot \sqrt{\iota} \tag{34}$$

$$\le 2c\beta(t)\eta r \cdot \sqrt{\iota}. \tag{35}$$

Moreover, due to assumption (28) and the Hessian Lipschitz property, we have

$$\|\Delta_t\| = \int_0^1 \nabla^2 F(y \cdot x_t + (1-y) \cdot x_t') \, \mathrm{d}y \tag{36}$$

$$\le \rho \max\{\|x_t - \tilde{x}\|, \|x_t' - \tilde{x}\|\} \le \rho\mathscr{S}. \tag{37}$$

With the above upper bounds on $\|\hat{x}_t\|$ and $\|\Delta_t\|$ for $t \le \tau$, we immediately get

$$\|\mathscr{P}_h(\tau + 1)\| \le \eta\rho\mathscr{S} \sum_{t=1}^{\tau+1} (1 + \eta\gamma)^{\tau+1-t} \left(2c \cdot \beta(t)\eta r\sqrt{\iota}\right) \tag{38}$$

$$\le 2\eta\rho\mathscr{S}\mathscr{T}c\beta(\tau+1)\eta r\sqrt{\iota} \le \frac{\beta(\tau+1)\eta r}{40}, \tag{39}$$

where the last inequality follows from $2c\eta\rho\mathscr{S}\mathscr{T} = \frac{2c}{s} \le \frac{1}{40}$ for large enough $s$ such that $s \ge 80c$.

Note that $\hat{\tilde{\zeta}}_t | \mathcal{F}_{t-1} \sim \mathrm{nSG}(M\|\hat{x}_t\|)$, by applying Lemma 12 with $B = [\alpha(t)]^2 \cdot \eta^2 M^2 \mathscr{S}^2$ and $b = [\alpha(t)]^2 \eta^2 M^2 \eta^2 r^2$ therein, we know that, with probability at least $1 - 4d\log\left(\frac{\mathscr{L}}{\eta r}\right) e^{-\iota}$, we have

$$\|\mathscr{P}_{sg}(\tau + 1)\| \le 2c\eta M\sqrt{\mathscr{T}}\beta(\tau)\eta r\sqrt{\iota}. \tag{40}$$

For large enough $s$ such that $s \ge (80c)^2$, we have $c\eta M\sqrt{\mathscr{T}\iota} \le \frac{2c}{\sqrt{s}} \le \frac{1}{40}$. Thus,

$$\|\mathscr{P}_{sg}(\tau + 1)\| \le c\eta M\sqrt{\mathscr{T}}\beta(\tau)\eta r\sqrt{\iota} \le \frac{\beta(\tau)\eta r}{40}. \tag{41}$$

By Lemma 18, we know that, when $t = \tau + 1$, with probability at least $1 - 2e^{-\iota}$, we have

$$\|\mathscr{P}_p(\tau + 1)\| \le c\beta(\tau+1)\eta r\sqrt{\iota} \tag{42}$$

By the union bound, with probability at least $1 - \left(6d\tau \log\left(\frac{\mathscr{L}}{\eta r}\right) e^{-\iota} + 4d\log\left(\frac{\mathscr{L}}{\eta r}\right) e^{-\iota} + 2e^{-\iota}\right) \ge 1 - 6d(\tau+1)\log\left(\frac{\mathscr{L}}{\eta r}\right) e^{-\iota}$,

$$\|\mathscr{P}_h(\tau + 1) + \mathscr{P}_{sg}(\tau + 1)\| \le \frac{\beta(\tau)\eta r}{20}, \qquad \|\mathscr{P}_p(\tau + 1)\| \le c\beta(\tau+1)\eta r\sqrt{\iota}, \tag{43}$$

which concludes the proof. $\qquad\qquad\square$

We continue the proof of Lemma 1. For large enough $\iota$ such that $\iota \ge \log\left(36d\mathscr{T}\log\left(\frac{\mathscr{L}}{\eta}\right)\right)$, which is promised by $\mu \ge \frac{1}{s}\log\left(\frac{9d}{C^{\frac{1}{4}}\eta\sqrt{s\rho\psi}}\log\left(\frac{4C^{\frac{1}{4}}}{s\eta r}\sqrt{\frac{\psi}{\rho}}\right)\right)$, we have $6d\mathscr{T}\log\left(\frac{\mathscr{L}}{\eta r}\right) e^{-\iota} \le \frac{1}{6}$. Then by Lemma 18 and Lemma 19, with probability at least $\frac{2}{3} - \frac{1}{6} = \frac{1}{2}$, we have

$$\|\mathscr{P}_p(\mathscr{T})\| \ge \frac{\beta(\mathscr{T})\eta r}{10}, \qquad \|\mathscr{P}_h(\mathscr{T}) + \mathscr{P}_{sg}(\mathscr{T})\| \le \frac{\beta(\mathscr{T})\eta r}{20}. \tag{44}$$

Combining (44) and the decomposition of $\hat{x}_t$ given by Lemma 17, we have

$$\max\left\{\|x_\mathscr{T} - \tilde{x}\|, \|x_\mathscr{T}' - \tilde{x}\|\right\} \tag{45}$$

$$\ge \frac{1}{2}\|\hat{x}_\mathscr{T}\| \ge \frac{1}{2}\left[\|\mathscr{P}_p(\mathscr{T})\| - \|\mathscr{P}_h(\mathscr{T}) + \mathscr{P}_{sg}(\mathscr{T})\|\right] \ge \frac{\beta(\mathscr{T})\eta r}{40} = \frac{(1 + \eta\gamma)^\mathscr{T}\sqrt{\eta r}}{40\sqrt{2}} \tag{46}$$

$$\ge \frac{(1 + \eta\sqrt{\rho\alpha})^\mathscr{T}\sqrt{\eta r}}{40\sqrt{2}} \ge \frac{2^{\eta\sqrt{\rho\alpha}\mathscr{T}}\sqrt{\eta r}}{40\sqrt{2}} = \frac{2^{\frac{\iota}{s}}\sqrt{\eta r}}{40\sqrt{2}} = \frac{2^\mu\sqrt{\eta r}}{40\sqrt{2}} > \mathscr{S}, \tag{47}$$

where the second last inequality is due to the fact $1 + a > 2^a, \forall a \in (0, 1]$ and $\eta\sqrt{\rho\alpha} \leq \frac{1}{\iota^2} \leq 1$, and the last inequality is because $\mu > \log\left(\frac{160\sqrt{2}C^{\frac{1}{4}}}{s\sqrt{\eta r}}\sqrt{\frac{\psi}{\rho}}\right)$. The above contradicts with our assumption (28). Thus, with probability at least $\frac{1}{2}$, $\exists t \leq \mathscr{T}, \max\{\|x_t - \tilde{x}\|^2, \|x'_t - \tilde{x}\|^2\} \geq \mathscr{S}^2$. $\qquad\square$

## D.2 PROOF OF LEMMA 2

*Proof of Lemma 2.* The failure probability after $Q$ repeats is at most $\left(\frac{3}{4}\right)^Q$, it is easy to verify that when $Q = \frac{5}{2}\log\frac{1}{\omega_0}$, $\left(\frac{3}{4}\right)^Q < \omega_0$, which concludes the proof. $\qquad\square$

## D.3 PROOF OF LEMMA 3

*Proof of Lemma 3.* For any $t > 1$, we have

$$F(x_t) - F(x_{t-1}) \leq \langle \nabla F(x_{t-1}), x_t - x_{t-1}\rangle + \frac{M}{2}\|x_t - x_{t-1}\|^2 \tag{48}$$

$$\leq -\eta\langle \nabla F(x_{t-1}), \hat{g}_{t-1}\rangle + \frac{M}{2}\eta^2\|\hat{g}_{t-1}\|^2 \tag{49}$$

$$\leq -\eta\langle \nabla F(x_{t-1}), \hat{g}_{t-1}\rangle + \frac{\eta}{2}\|\hat{g}_{t-1}\|^2 \tag{50}$$

$$\leq \frac{\eta}{2}\|\nu_t\|^2 - \frac{\eta}{2}\|\nabla F(x_{t-1})\|^2 - \frac{\eta}{2}\|\hat{g}_{t-1}\|^2 + \frac{\eta}{2}\|\hat{g}_{t-1}\|^2 \tag{51}$$

$$= -\frac{\eta}{2}\|\nabla F(x_{t-1})\|^2 + \frac{\eta}{2}\|\nu_t\|^2. \tag{52}$$

Thus, for any time step $t_0$, we have

$$F(x_{t_0+t}) - F(x_{t_0}) \leq -\frac{\eta}{2}\sum_{i=0}^{t-1}\|\nabla F(x_{t_0+i})\|^2 + \frac{\eta}{2}\sum_{i=1}^{t}\|\nu_{t_0+i}\|^2 \tag{53}$$

$$\square$$

## D.4 PROOF OF COROLLARY 2

*Proof of Corollary 2.* Note that

$$\frac{\eta}{2}\sum_{i=1}^{t}\|\nu_{t_0+i}\|^2 = \frac{\eta}{2}\sum_{i=1}^{t}\|\hat{\zeta}_{t_0+i} + \hat{\xi}_{t_0+i}\|^2 \leq \eta\sum_{i=1}^{t}(\|\hat{\zeta}_{t_0+i}\|^2 + \|\hat{\xi}_{t_0+i}\|^2) \tag{54}$$

Since $\hat{\zeta}_i \sim \mathrm{nSG}(\sigma), \forall i$, by Lemma 13, with probability at least $1 - e^{-\iota}$, $\sum_{i=1}^{t}\|\hat{\zeta}_{t_0+i}\|^2 \leq C \cdot \sigma^2(t + \iota)$. On the other hand, $\hat{\xi}_i \sim \mathrm{SG}(r), \forall i$, by Lemma 10, we know $\hat{\xi}_i \sim \mathrm{nSG}(2\sqrt{2} \cdot r\sqrt{d})$. By Lemma 13 again, with probability at least $1 - e^{-\iota}$, $\sum_{i=1}^{t}\|\hat{\xi}_{t_0+i}\|^2 \leq 8C \cdot r^2 d(t+\iota)$. Combine the two upper bounds and apply the union bound, we get the desired upper bound on $\frac{\eta}{2}\sum_{i=1}^{t}\|\nu_{t_0+i}\|^2$. $\qquad\square$

## D.5 PROOF OF LEMMA 4

*Proof of Lemma 4.* Note that,

$$\|x_{t_0+\tau} - x_{t_0}\|^2 = \eta^2\left\|\sum_{t=1}^{\tau}\nabla F(x_{t_0+t-1}) + \nu_{t_0+t}\right\|^2 \tag{55}$$

$$\leq 2\eta^2\left[\left\|\sum_{t=1}^{\tau}\nabla F(x_{t_0+t-1})\right\|^2 + \left\|\sum_{t=1}^{\tau}\nu_{t_0+t}\right\|^2\right] \tag{56}$$

$$\leq 2\eta^2\tau\sum_{t=1}^{\tau}\|\nabla F(x_{t_0+t-1})\|^2 + 2\eta^2\tau\sum_{t=1}^{\tau}\|\nu_{t_0+t}\|^2 \tag{57}$$

By following the same argument as in the proof of corollary 2, with probability at least $1 - 2e^{-\iota}$,

$$\sum_{t=1}^{\tau} \|\nu_{t_0+t}\| \le c \cdot \psi^2(\tau + \iota). \tag{58}$$

Again by corollary 2, with the same probability,

$$\sum_{t=1}^{\tau} \|\nabla F(x_{t_0+t-1})\|^2 \le \frac{2}{\eta}[F(x_{t_0}) - F(x_{t_0+\tau})] + c \cdot \psi^2(\tau + \iota). \tag{59}$$

Therefore, with probability at least $1 - 2e^{-\iota}$,

$$\|x_{t_0+\tau} - x_{t_0}\|^2 \le 4\eta\tau[F(x_{t_0}) - F(x_{t_0+\tau})] + 4c \cdot \eta^2\tau\psi^2(\tau + \iota). \tag{60}$$

By re-arranging terms above, we have

$$F(x_{t_0+\tau}) - F(x_{t_0}) \le -\frac{1}{4\eta\tau}\|x_{t_0+\tau} - x_{t_0}\|^2 + c \cdot \eta\psi^2(\tau + \iota). \tag{61}$$

According to our criterion for successful escape, we know that $\|x_{t_0+\tau} - x_{t_0}\| \ge \mathscr{S}$. Then

$$F(x_{t_0+\tau}) - F(x_{t_0}) \le -\frac{1}{4\eta\tau}\|x_{t_0+\tau} - x_{t_0}\|^2 + c \cdot \eta\psi^2(\tau + \iota) \tag{62}$$

$$\le -\frac{\mathscr{S}^2}{4\eta\mathscr{T}} + c \cdot \eta\psi^2(\mathscr{T} + \iota) \tag{63}$$

$$\le -\frac{s}{4\iota^3}\sqrt{\frac{\alpha^3}{\rho}} + \frac{2c \cdot \psi^2\iota}{s\sqrt{\rho\alpha}} \tag{64}$$

$$\le -\frac{s}{8\iota^3}\sqrt{\frac{\alpha^3}{\rho}} = \mathscr{F}, \tag{65}$$

where the second to last inequality is from the fact that $s\eta\sqrt{\rho\alpha} = \frac{\rho\alpha}{M^2 s\mu^2} < 1$, and the last inequality follows from $\alpha \ge 4\sqrt{C}s\mu^2\psi$. $\qquad\square$

### D.6 PROOF OF LEMMA 5

*Proof of Lemma 5.* By corollary 3, we know that for $\forall t$, $\nu_t \sim \mathrm{nSG}(C\sqrt{\sigma^2 + r^2 d})$. Since $\mathbb{E}[\nu_t] = 0$, according to definition 6, with probability at least $1 - \frac{\omega}{2T}$

$$\|\nu_t\| \le \sqrt{2}C \cdot \psi\sqrt{\log\frac{4T}{\omega}} \le \chi. \tag{66}$$

Applying the union bound to above immediately guarantees that, for all $t \in [T]$, with probability at least $1 - \frac{\omega}{2}$, the gradient estimation error $\|\hat{g}_t - \nabla F(x_{t-1})\|$ does not exceed $\chi$. $\qquad\square$

### D.7 PROOF OF LEMMA 6

*Proof of Lemma 6.* By Lemma 5, for all $t \in [T]$, with probability at least $1 - \frac{\omega}{2}$, the gradient estimation error $\|\hat{g}_t - \nabla F(x_{t-1})\|$ does not exceed $\chi$ defined in (4). With this hold, we know that, for any time step $t \in [T]$, if $\|\hat{g}_t\| \le 3\chi$ so that the escape process is triggered, we know the true gradient norm $\|\nabla F(x_{t-1})\|$ is at most $\alpha = 4\chi$, otherwise, when the algorithm runs out of the escape process, the true gradient norm $\|\nabla F(x_{t-1})\|$ is at least $2\chi$. With this insight, we discuss the average function value decrease per PSGD step for above two cases separately.

- **Case 1.** When the algorithm runs in the escape process, the average decrease in the function value during the successful `Escape` process is

$$\frac{\mathscr{F}}{\mathscr{T}} = \frac{s^2\alpha^2\eta}{8\iota^4} = \frac{2\chi^2\eta}{s^2\mu^4}. \tag{67}$$

- **Case 2.** When the algorithm runs out of the escape process, for every PSGD update, the function value is decreased by at least

$$\frac{\eta}{2}(2\chi)^2 = 2\chi^2\eta > \frac{2\chi^2\eta}{s^2\mu^4}. \tag{68}$$

Combining the two cases, the total number of **effective** descent steps is at most

$$T_{\text{effective}} := \frac{(F_0 - F^*)s^2\mu^4}{2\chi^2\eta}. \tag{69}$$

The total number of strict saddle points we need to escape is at most:

$$N_{\text{saddle}} := \frac{F_0 - F^*}{\mathscr{F}} = \frac{8\iota^3(F_0 - F^*)}{s}\sqrt{\frac{\rho}{\chi^3}}. \tag{70}$$

We know that, at each $\alpha$-saddle point, we escape it successfully with probability at least $\frac{1}{4}$. To boost the success probability for all such escapes to desired $1 - \frac{\omega}{2}$, we need to repeat the process for $Q$ times independently. By applying Lemma 2 with $\omega_0 = \frac{\omega}{2N_{\text{saddle}}}$ therein, $Q$ should be set as

$$Q = \frac{5}{2}\log\left(\frac{16\iota^3(F_0 - F^*)}{s\omega}\sqrt{\frac{\rho}{\chi^3}}\right). \tag{71}$$

Therefore, the maximum total number of all descent steps the algorithm performs (sequential and parallel) is bounded as

$$T \leq T_{\text{effective}} \cdot Q = \frac{5(F_0 - F^*)s^2\mu^4}{4\chi^2\eta}\log\left(\frac{16\iota^3(F_0 - F^*)}{s\omega}\sqrt{\frac{\rho}{\chi^3}}\right) = \tilde{O}\left(\frac{U}{\eta\chi^2}\right). \tag{72}$$

$\square$

# E   OMITTED PROOFS IN SECTION 5

## E.1   PROOF OF LEMMA 7

*Proof of Lemma 7.* For $\forall t$, let $\tau(t)$ be the last iteration till $t$ when $\mathcal{O}_1$ was used. If $t = \tau(t)$, then we have

$$\hat{g}_t = \mathcal{O}_1(x_{t-1}, \mathcal{B}_t) + \xi_t. \tag{73}$$

$\mathcal{O}_1(x_{t-1}, \mathcal{B}_t)$ is an unbiased estimate of $\nabla F(x_{t-1})$. Denote $\zeta_t := \mathcal{O}_1(x_{t-1}, \mathcal{B}_t) - \nabla F(x_{t-1})$, then

$$\hat{g}_t - \nabla F(x_{t-1}) = \zeta_t + \xi_t, \tag{74}$$

By the $G$-Lipschitzness of loss function $f$, $\zeta_t \sim \text{nSG}\left(\frac{G\sqrt{\log d}}{\sqrt{b_1}}\right)$. According to the algorithm, $\xi_t$ is also zero-mean and $\xi_t \sim \mathcal{N}\left(0, c_1\frac{G^2\log\frac{1}{\delta}}{b_1^2\epsilon^2}\mathbf{I}_d\right)$. In this case, the lemma holds.

If $t > \tau(t)$, then we have

$$\hat{g}_t = \mathcal{O}_1(x_{\tau(t)-1}, \mathcal{B}_{\tau(t)}) + \xi_{\tau(t)} + \sum_{i=\tau(t)+1}^{t}\left(\mathcal{O}_2(x_{i-1}, x_{i-2}, \mathcal{B}_i) + \xi_i\right), \tag{75}$$

$\mathcal{O}_2(x_{i-1}, x_{i-2}, \mathcal{B}_i)$ is an unbiased estimate of $\nabla F(x_{i-1}) - \nabla F(x_{i-2})$. Denote $\zeta'_i := \mathcal{O}_2(x_{i-1}, x_{i-2}, \mathcal{B}_i) - [\nabla F(x_{i-1}) - \nabla F(x_{i-2})]$, then

$$\hat{g}_t - \nabla F(x_{t-1}) = \hat{g}_t - \left(\nabla F(x_{\tau(t)-1}) + \sum_{i=\tau(t)+1}^{t}[\nabla F(x_{i-1}) - \nabla F(x_{i-2})]\right) \tag{76}$$

$$= \zeta_{\tau(t)} + \sum_{i=\tau(t)+1}^{t}\zeta'_i + \xi_{\tau(t)} + \sum_{i=\tau(t)+1}^{t}\xi'_i, \tag{77}$$

By the $M$-smoothness of loss function $f$, each $\zeta_i' \sim \text{nSG}\left(\frac{M\|x_{i-1}-x_{i-2}\|\sqrt{\log d}}{\sqrt{b_2}}\right)$. According to the algorithm, $\xi_i' \sim \mathcal{N}\left(0, c_2\frac{M^2\log\frac{1}{\delta}}{b_2^2\epsilon^2}\|x_{t-1}-x_{t-2}\|^2\mathbf{I}_d\right)$. By corollary 3 and the fact ensured by our algorithm that, $\text{drift}_t = \sum_{i=\tau(t)+1}^{t}\|x_{i-1}-x_{i-2}\|^2 \leq \kappa$ almost surely, we have

$$\sigma \leq O\left(\sqrt{\left[\left(\frac{G\sqrt{\log d}}{\sqrt{b_1}}\right)^2 + \sum_{i=\tau(t)+1}^{t}\left(\frac{M\|x_{i-1}-x_{i-2}\|\sqrt{\log d}}{\sqrt{b_2}}\right)^2\right]\cdot\log d}\right) \tag{78}$$

$$\leq O\left(\sqrt{\frac{G^2\log^2 d}{b_1} + \frac{M^2\log^2 d}{b_2}\kappa}\right). \tag{79}$$

By the properties of Gaussian distribution, and the fact that, $\text{drift}_t = \sum_{i=\tau(t)+1}^{t}\|x_{i-1}-x_{i-2}\|^2 \leq \kappa$ almost surely, we have

$$r \leq O\left(\sqrt{\frac{G^2\log\frac{1}{\delta}}{b_1^2\epsilon^2} + \sum_{i=\tau(t)+1}^{t}\left(\frac{M^2\log\frac{1}{\delta}}{b_2^2\epsilon^2}\|x_{t-1}-x_{t-2}\|^2\right)}\right) \tag{80}$$

$$\leq O\left(\sqrt{\frac{G^2\log\frac{1}{\delta}}{b_1^2\epsilon^2} + \frac{M^2\log\frac{1}{\delta}}{b_2^2\epsilon^2}\kappa}\right). \tag{81}$$

$\square$

### E.2 PROOF OF LEMMA 8

*Proof of Lemma 8.* By $\eta \leq \frac{1}{M}$ and $M$-smoothness, we have

$$F(x_t) - F(x_{t-1}) \leq \langle\nabla F(x_{t-1}), x_t - x_{t-1}\rangle + \frac{M}{2}\|x_t - x_{t-1}\|^2$$

$$\leq \langle\nabla F(x_{t-1}) - \hat{g}_t, -\eta\cdot\hat{g}_t\rangle - \eta\|\hat{g}_t\|^2 + \frac{\eta}{2}\|\hat{g}_t\|^2$$

$$\leq \eta\|\nabla F(x_{t-1}) - \hat{g}_t\|\|\hat{g}_t\|_2 - \frac{\eta}{2}\|\hat{g}_t\|^2.$$

By Lemma 5, we know that, with probability at least $1 - \frac{\omega}{2}$, the gradient estimation error $\|\hat{g}_t - \nabla F(x_{t-1})\| = \|\nu_t\| \leq \chi$ holds for all $t \in [T]$. If $\|\nabla F(x_{t-1})\| \geq 4\chi$, then we have

$$\|\hat{g}_t\| \geq 3\chi \geq 3\|\nabla F(x_{t-1}) - \hat{g}_t\|,$$

which further leads to

$$F(x_t) - F(x_{t-1}) \leq -\frac{\eta}{6}\|\hat{g}_t\|^2.$$

If $\|\nabla F(x_{t-1})\|_2 \leq 4\chi$, then $\|\hat{g}_t\| \leq 5\chi$, which implies

$$F(x_t) - F(x_{t-1}) \leq 5\eta\chi^2.$$

Index the items in $\mathcal{T}$ with $\mathcal{T} = \{t_1, \cdots, t_{|\mathcal{T}|}\}$ such that $t_i < t_{i+1}, \forall 1 \leq i \leq |\mathcal{T}| - 1$. Then

$$F(x_{t_{i+1}}) - F(x_{t_i}) \leq -\frac{1}{6\eta}\sum_{t=t_i+1}^{t_{i+1}}\eta^2\|\hat{g}_t\|_2^2 + (t_{i+1}-t_i)5\eta\chi^2$$

$$\leq -\frac{1}{6\eta}\text{drift}_{t_{i+1}} + (t_{i+1}-t_i)5\eta\chi^2 \leq -\frac{1}{6\eta}\kappa + (t_{i+1}-t_i)5\eta\chi^2.$$

Summing over all indices, we have

$$F(x_{t_{|\mathcal{T}|}}) - F(x_{t_1}) \leq -\frac{|\mathcal{T}|}{6\eta}\kappa + 5T\eta\chi^2.$$

Since the risk function is upper bounded by $U$, there must be $F(x_{t_{|\mathcal{T}|}}) - F(x_{t_1}) \geq -U$, which gives

$$|\mathcal{T}| \leq O\left(\frac{U\eta}{\kappa} + \frac{T\eta^2\chi^2}{\kappa}\right) = O\left(\frac{U\eta}{\kappa}\right).$$

$\square$

### E.3 PROOF OF THEOREM 2

*Proof of Theorem 2.* We first show that, our setting of $b_1$ and $b_2$ is feasible. Specifically, we need to show the total sample used by each client is $O(n)$. This can be verified as follows.

$$b_1 \cdot |\mathcal{T}| + b_2 \cdot (T - |\mathcal{T}|) \le b_1 \cdot |\mathcal{T}| + b_2 \cdot T \le O(n),$$

where we use the fact $T = O\left(\frac{U}{\eta\chi^2}\right)$ and $|\mathcal{T}| \le O(\frac{U\eta}{\kappa})$ given in Lemma 8.

As we never reuse a sample, the privacy guarantee follows directly from the Gaussian mechanism and the parallel composition property.

By Theorem 1,

$$\alpha = O(\chi) = \tilde{O}(\psi) \le \tilde{O}(\sqrt{\sigma^2 + r^2 d})$$

$$= \tilde{O}\left(\sqrt{\frac{G^2}{b_1} + \frac{G^2 d}{b_1^2 \epsilon^2} + \left(\frac{M^2}{b_2} + \frac{M^2 d}{b_2^2 \epsilon^2}\right) \cdot \kappa}\right).$$

By our setting of $b_1 = \frac{n\kappa}{2U\eta}$ and $b_2 = \frac{n\eta\chi^2}{2U}$, we further have

$$\alpha = \tilde{O}\left(\sqrt{\frac{G^2 U\eta}{n\kappa} + \frac{G^2 d U^2 \eta^2}{n^2 \epsilon^2 \kappa^2} + \frac{M^2 U\kappa}{n\eta\chi^2} + \frac{M^2 d U^2 \kappa}{n^2 \epsilon^2 \eta^2 \chi^4}}\right) \tag{82}$$

$$= \tilde{O}\left(\sqrt{\frac{G^2 U\rho^{\frac{1}{2}}\alpha^{\frac{1}{2}}}{M^2 n\kappa} + \frac{G^2 d U^2 \rho\alpha}{n^2 \epsilon^2 M^4 \kappa^2} + \frac{M^4 U\kappa}{n\rho^{\frac{1}{2}}\alpha^{\frac{5}{2}}} + \frac{M^6 d U^2 \kappa}{n^2 \epsilon^2 \rho\alpha^5}}\right), \tag{83}$$

which gives us that

$$\alpha = \tilde{O}\left(\max\left\{\left(\frac{G^2 U\sqrt{\rho}}{M^2 n\kappa}\right)^{\frac{2}{3}}, \frac{G^2 d U^2 \rho}{n^2 \epsilon^2 M^4 \kappa^2}, \left(\frac{M^4 U\kappa}{n\sqrt{\rho}}\right)^{\frac{2}{9}}, \left(\frac{M^6 d U^2 \kappa}{n^2 \epsilon^2 \rho}\right)^{\frac{1}{7}}\right\}\right).$$

Setting $\kappa = \max\left\{\frac{G^{\frac{3}{2}} U^{\frac{1}{2}} \rho^{\frac{1}{2}}}{M^{\frac{5}{2}} n^{\frac{1}{2}}}, \frac{G^{\frac{14}{15}} d^{\frac{2}{5}} U^{\frac{4}{5}} \rho^{\frac{8}{15}}}{M^{\frac{34}{15}} (n\epsilon)^{\frac{4}{5}}}\right\}$, we get

$$\alpha = \tilde{O}\left(\left(\frac{GUM}{n}\right)^{\frac{1}{3}} + \frac{G^{\frac{2}{15}} U^{\frac{2}{5}} M^{\frac{8}{15}}}{\rho^{\frac{1}{15}}} \left(\frac{\sqrt{d}}{n\epsilon}\right)^{\frac{2}{5}}\right) = \tilde{O}\left(\frac{1}{n^{\frac{1}{3}}} + \left(\frac{\sqrt{d}}{n\epsilon}\right)^{\frac{2}{5}}\right).$$

$\square$

## F OMITTED PROOFS IN SECTION 6

### F.1 PROOF OF LEMMA 9

*Proof of Lemma 9.* For $\forall t$, let $\tau(t)$ be the last iteration till $t$ when $\mathcal{O}_1$ was used. If $t = \tau(t)$, then we have

$$\hat{g}_t = \frac{1}{m} \sum_{j=1}^{m} (\mathcal{O}_1(x_{t-1}, \mathcal{B}_{j,t}) + \xi_{j,t}). \tag{84}$$

$\mathcal{O}_1(x_{t-1}, \mathcal{B}_{j,t})$ is an unbiased estimate of $\nabla F_j(x_{t-1})$. Denote $\zeta_{j,t} := \mathcal{O}_1(x_{t-1}, \mathcal{B}_{j,t}) - \nabla F_j(x_{t-1})$, $\zeta_t := \frac{1}{m} \sum_{j=1}^{m} \zeta_{j,t}$ and $\xi_t := \frac{1}{m} \sum_{j=1}^{m} \xi_{j,t}$, then

$$\hat{g}_t - \nabla F(x_{t-1}) = \frac{1}{m} \sum_{j=1}^{m} (\hat{g}_{j,t} - \nabla F_j(x_{t-1})) = \frac{1}{m} \sum_{j=1}^{m} (\zeta_{j,t} + \xi_{j,t}) = \zeta_t + \xi_t, \tag{85}$$

By the $G$-Lipschitzness of loss function $f$, $\zeta_t \sim \text{nSG}\left(\frac{G\sqrt{\log d}}{\sqrt{m \cdot b_1}}\right)$. According to the algorithm, each $\xi_{j,t}$ is zero-mean and $\xi_{j,t} \sim \mathcal{N}\left(0, c_1 \frac{G^2 \log \frac{1}{\delta}}{b_1^2 \epsilon^2} \mathbf{I}_d\right)$, and thus $\xi_t \sim \mathcal{N}\left(0, c_1 \frac{G^2 \log \frac{1}{\delta}}{m \cdot b_1^2 \epsilon^2} \mathbf{I}_d\right)$. In this case, the lemma holds.

If $t > \tau(t)$, then we have

$$\hat{g}_t = \frac{1}{m} \sum_{j=1}^{m} \left[ \mathcal{O}_1(x_{\tau(t)-1}, \mathcal{B}_{j,\tau(t)-1}) + \xi_{j,\tau(t)} + \sum_{i=\tau(t)+1}^{t} \left( \mathcal{O}_2(x_{i-1}, x_{i-2}, \mathcal{B}_{j,i}) + \xi'_{j,i} \right) \right], \quad (86)$$

Each $\mathcal{O}_2(x_{i-1}, x_{i-2}, \mathcal{B}_{j,i})$ is an unbiased estimate of $\nabla F_j(x_{i-1}) - \nabla F_j(x_{i-2})$. Denote $\zeta'_{j,i} :=$ $\mathcal{O}_2(x_{i-1}, x_{i-2}, \mathcal{B}_{j,i}) - [\nabla F_j(x_{i-1}) - \nabla F_j(x_{i-2})]$, $\zeta'_i := \frac{1}{m} \sum_{j=1}^{m} \zeta'_{j,i}$ and $\xi'_i := \frac{1}{m} \sum_{j=1}^{m} \xi'_{j,i}$ then

$$\hat{g}_t - \nabla F(x_{t-1}) = \frac{1}{m} \sum_{j=1}^{m} \left[ \hat{g}_{j,t} - \left( \nabla F_j(x_{\tau(t)-1}) + \sum_{i=\tau(t)+1}^{t} [\nabla F_j(x_{i-1}) - \nabla F_j(x_{i-2})] \right) \right] \tag{87}$$

$$= \frac{1}{m} \sum_{j=1}^{m} \left[ \zeta_{j,\tau(t)} + \sum_{i=\tau(t)+1}^{t} \zeta'_{j,i} + \xi_{j,\tau(t)} + \sum_{i=\tau(t)+1}^{t} \xi'_{j,i} \right] \tag{88}$$

$$= \zeta_{\tau(t)} + \sum_{i=\tau(t)+1}^{t} \zeta'_i + \xi_{\tau(t)} + \sum_{i=\tau(t)+1}^{t} \xi'_i \tag{89}$$

By the $M$-smoothness of loss function $f$, each $\zeta'_i \sim \text{nSG}\left( \frac{M\|x_{i-1}-x_{i-2}\|\sqrt{\log d}}{\sqrt{m \cdot b_2}} \right)$. According to the algorithm, each $\xi'_{j,i} \sim \mathcal{N}\left( 0, c_2 \frac{M^2 \log \frac{1}{\delta}}{b_2^2 \epsilon^2} \|x_{t-1} - x_{t-2}\|^2 \mathbf{I}_d \right)$, thus $\xi'_i \sim \mathcal{N}\left( 0, c_2 \frac{M^2 \log \frac{1}{\delta}}{m \cdot b_2^2 \epsilon^2} \|x_{t-1} - x_{t-2}\|^2 \mathbf{I}_d \right)$. By corollary 3 and the fact ensured by our algorithm that, $\text{drift}_t = \sum_{i=\tau(t)+1}^{t} \|x_{i-1} - x_{i-2}\|^2 \leq \kappa$ almost surely, we have

$$\sigma \leq O\left( \sqrt{ \left[ \left( \frac{G\sqrt{\log d}}{\sqrt{m \cdot b_1}} \right)^2 + \sum_{i=\tau(t)+1}^{t} \left( \frac{M\|x_{i-1} - x_{i-2}\|\sqrt{\log d}}{\sqrt{m \cdot b_2}} \right)^2 \right] \cdot \log d } \right) \tag{90}$$

$$\leq O\left( \sqrt{ \frac{G^2 \log^2 d}{m \cdot b_1} + \frac{M^2 \log^2 d}{m \cdot b_2} \kappa } \right). \tag{91}$$

By the properties of Gaussian distribution, and the fact that, $\text{drift}_t = \sum_{i=\tau(t)+1}^{t} \|x_{i-1} - x_{i-2}\|^2 \leq \kappa$ almost surely, we have

$$r \leq O\left( \sqrt{ \frac{G^2 \log \frac{1}{\delta}}{m \cdot b_1^2 \epsilon^2} + \sum_{i=\tau(t)+1}^{t} \left( \frac{M^2 \log \frac{1}{\delta}}{m \cdot b_2^2 \epsilon^2} \|x_{t-1} - x_{t-2}\|^2 \right) } \right) \tag{92}$$

$$\leq O\left( \sqrt{ \frac{G^2 \log \frac{1}{\delta}}{m \cdot b_1^2 \epsilon^2} + \frac{M^2 \log \frac{1}{\delta}}{m \cdot b_2^2 \epsilon^2} \kappa } \right). \tag{93}$$

$\square$

### F.2 PROOF OF THEOREM 3

*Proof of Theorem 3.* We first show that, our setting of $b_1$ and $b_2$ is feasible. Specifically, we need to show the total sample used by each client is $O(n)$. This can be verified as follows.

$$b_1 \cdot |\mathcal{T}| + b_2 \cdot (T - |\mathcal{T}|) \leq b_1 \cdot |\mathcal{T}| + b_2 \cdot T \leq O(n),$$

where we use the fact $T = O\left( \frac{U}{\eta \chi^2} \right)$ and $|\mathcal{T}| \leq O(\frac{U\eta}{\kappa})$ given in Lemma 8.

As we never reuse a sample, the privacy guarantee follows directly from the Gaussian mechanism and the parallel composition property.

By Theorem 1,

$$\alpha = O(\chi) = \tilde{O}(\psi) \le \tilde{O}(\sqrt{\sigma^2 + r^2 d})$$

$$= \tilde{O}\left(\sqrt{\frac{G^2}{m \cdot b_1} + \frac{G^2 d}{m \cdot b_1^2 \epsilon^2} + \left(\frac{M^2}{m \cdot b_2} + \frac{M^2 d}{m \cdot b_2^2 \epsilon^2}\right) \cdot \kappa}\right).$$

By our setting of $b_1 = \frac{n\kappa}{2U\eta}$ and $b_2 = \frac{n\eta\chi^2}{2U}$, we further have

$$\alpha = \tilde{O}\left(\sqrt{\frac{G^2 U\eta}{m \cdot n\kappa} + \frac{G^2 dU^2\eta^2}{m \cdot n^2\epsilon^2\kappa^2} + \frac{M^2 U\kappa}{m \cdot n\eta\chi^2} + \frac{M^2 dU^2\kappa}{m \cdot n^2\epsilon^2\eta^2\chi^4}}\right) \tag{94}$$

$$= \tilde{O}\left(\sqrt{\frac{G^2 U\rho^{\frac{1}{2}}\alpha^{\frac{1}{2}}}{m \cdot M^2 n\kappa} + \frac{G^2 dU^2\rho\alpha}{m \cdot n^2\epsilon^2 M^4\kappa^2} + \frac{M^4 U\kappa}{m \cdot n\rho^{\frac{1}{2}}\alpha^{\frac{5}{2}}} + \frac{M^6 dU^2\kappa}{m \cdot n^2\epsilon^2\rho\alpha^5}}\right), \tag{95}$$

which gives us that

$$\alpha = \tilde{O}\left(\max\left\{\left(\frac{G^2 U\sqrt{\rho}}{m \cdot M^2 n\kappa}\right)^{\frac{2}{3}}, \frac{G^2 dU^2\rho}{m \cdot n^2\epsilon^2 M^4\kappa^2}, \left(\frac{M^4 U\kappa}{m \cdot n\sqrt{\rho}}\right)^{\frac{2}{9}}, \left(\frac{M^6 dU^2\kappa}{m \cdot n^2\epsilon^2\rho}\right)^{\frac{1}{7}}\right\}\right).$$

Setting $\kappa = \max\left\{\frac{G^{\frac{3}{2}}U^{\frac{1}{2}}\rho^{\frac{1}{2}}}{M^{\frac{5}{2}}(mn)^{\frac{1}{2}}}, \frac{G^{\frac{14}{15}}d^{\frac{2}{5}}U^{\frac{4}{5}}\rho^{\frac{8}{15}}}{M^{\frac{34}{15}}(\sqrt{mn}\epsilon)^{\frac{4}{5}}}\right\}$, we get

$$\alpha = \tilde{O}\left(\left(\frac{GUM}{mn}\right)^{\frac{1}{3}} + \frac{G^{\frac{2}{15}}U^{\frac{2}{5}}M^{\frac{8}{15}}}{\rho^{\frac{1}{15}}}\left(\frac{\sqrt{d}}{\sqrt{mn}\epsilon}\right)^{\frac{2}{5}}\right) = \tilde{O}\left(\frac{1}{(mn)^{\frac{1}{3}}} + \left(\frac{\sqrt{d}}{\sqrt{mn}\epsilon}\right)^{\frac{2}{5}}\right).$$

$\square$

### F.3 PROOF OF THEOREM 4

*Proof of Theorem 4.* Define $\mathcal{S} := \bigsqcup_{j=1}^m S_j$. Define $\theta_p := \frac{1}{m}\sum_{j=1}^m \theta_{j,p}$ and $\mathbf{H}_p := \frac{1}{m}\sum_{j=1}^m \mathbf{H}_{j,p}$. Denote $\sigma_1^2 = c_1 \frac{G^2 T \log\frac{1}{\delta}}{n^2\epsilon^2}$ and $\sigma_2^2 = c_2 \frac{M^2 dT \log\frac{1}{\delta}}{n^2\epsilon^2}$ for simplicity.

For any $\mathcal{S}_j$ and $x$, $\nabla f(x; \mathcal{S}_j) - \nabla F_j(x)$ is zero-mean and follows $\text{nSG}\left(\frac{2G}{\sqrt{n}}\right)$. By the Hoeffding inequality for norm-sub-Gaussian (Lemma 11), with probability at least $1 - \frac{\omega'}{8}$, we have

$$\|\nabla F(x_p) - \nabla f(x_p; \mathcal{S})\|_2 \le O\left(\frac{G\sqrt{\log\left(\frac{d}{\omega'}\right)}}{\sqrt{mn}}\right). \tag{96}$$

Note that $\theta_p \sim \mathcal{N}(0, \frac{\sigma_1^2}{m})$. By Lemma 10 and Definition 6, with probability at least $1 - \frac{\omega'}{8}$, we have

$$\|\xi_p\|_2 \le \sqrt{\frac{2\sigma_1^2 \log\left(\frac{16}{\omega'}\right)}{m}} = \frac{G\sqrt{2c_1 dT \log\left(\frac{1}{\delta}\right)\log\left(\frac{16}{\omega'}\right)}}{\sqrt{mn}\epsilon} = O\left(\frac{G\sqrt{dT \log\left(\frac{1}{\delta}\right)\log\left(\frac{1}{\omega'}\right)}}{\sqrt{mn}\epsilon}\right). \tag{97}$$

For any $j \in [m]$ and $z \in \mathcal{S}_j$, $\mathbb{E}[\nabla^2 f(x_p; z) - \nabla^2 F_j(x_p)] = 0$, and $\|\nabla^2 f(x_p; z) - \nabla^2 F_j(x_p)\|_2 \le 2M$ (due to $M$-smoothness). Applying Matrix Bernstein inequality (Lemma 14) with $\sigma^2 = 4M^2 mn$ and $t = 4M\sqrt{mn \log\frac{8d}{\omega'}}$ therein and noting that $mn \ge \frac{4}{9}\log\frac{8d}{\omega'}$, we have

$$\mathbb{P}\left[\left\|\sum_{j \in [m]}\left(\sum_{z \in \mathcal{S}_j}\nabla^2 f(x_p; z) - \nabla^2 F_j(x_p)\right)\right\|_2 \ge 4M\sqrt{mn \log\frac{8d}{\omega'}}\right] \le \frac{\omega'}{8}. \tag{98}$$

Therefore, with probability at least $1 - \frac{\omega'}{8}$, we have

$$\left\|\nabla^2 f(x_p; \mathcal{S}) - \nabla^2 F(x_p)\right\|_2 \le 4M\sqrt{\frac{\log \frac{8d}{\omega'}}{mn}} \le O\left(M\sqrt{\frac{\log \frac{d}{\omega'}}{mn}}\right). \tag{99}$$

Note that each on-and-above-diagonal entry in $\mathbf{H}_p$ samples from $\mathcal{N}(0, \frac{\sigma_2^2}{m})$. By Lemma 15, with probability at least $1 - \frac{\omega'}{8}$, we have

$$\|\mathbf{H}_p\|_2 \le O\left(\frac{\sigma_2}{\sqrt{m}}\left(\sqrt{d} + \sqrt{\log\left(\frac{32}{\omega'}\right)}\right)\right) = O\left(\frac{Md\sqrt{T\log\frac{1}{\delta}}}{\sqrt{m}n\epsilon} + \frac{M\sqrt{dT\log\left(\frac{1}{\delta}\right)\log\left(\frac{32}{\omega'}\right)}}{\sqrt{m}n\epsilon}\right) \tag{100}$$

$$\le O\left(\frac{Md\sqrt{T\log\frac{1}{\delta}\log\left(\frac{32}{\omega'}\right)}}{\sqrt{m}n\epsilon}\right). \tag{101}$$

Combining the results above, with probability at least $1 - \frac{\omega'}{2}$, we have

$$\|\nabla \bar{F}(x_p)\|_2 \le \|\nabla \bar{F}(x_p) - \nabla F(x_p)\|_2 + \|\nabla F(x_p)\|_2 \tag{102}$$

$$\le \|\nabla f(x_p; \mathcal{S}) - \nabla F(x_p)\|_2 + \|\xi_p\|_2 + \|\nabla F(x_p)\|_2 \tag{103}$$

$$\le O\left(\frac{G\log\left(\frac{8d}{\omega'}\right)}{\sqrt{mn}} + \frac{G\sqrt{dT\log\left(\frac{1}{\delta}\right)\log\left(\frac{16}{\omega'}\right)}}{\sqrt{m}n\epsilon} + \alpha\right), \tag{104}$$

and

$$\lambda_{\min}\left(\nabla^2 \bar{F}(x_p)\right) \ge \lambda_{\min}\left(\nabla^2 \bar{F}(x_p) - \nabla^2 F(x_p)\right) + \lambda_{\min}\left(\nabla^2 F(x_p)\right) \tag{105}$$

$$\ge \lambda_{\min}\left(\nabla^2 f(x_p; \mathcal{S}) - \nabla^2 F(x_p)\right) + \lambda_{\min}\left(\mathbf{H}_p\right) + \lambda_{\min}\left(\nabla^2 F(x_p)\right) \tag{106}$$

$$\ge -\left\|\nabla^2 f(x_p; \mathcal{S}) - \nabla^2 F(x_p)\right\|_2 - \|\mathbf{H}_p\|_2 - \sqrt{\rho\alpha} \tag{107}$$

$$\ge -O\left(M\sqrt{\frac{\log\left(\frac{8d}{\omega'}\right)}{mn}} + \frac{Md\sqrt{T\log\frac{1}{\delta}\log\left(\frac{32}{\omega'}\right)}}{\sqrt{m}n\epsilon} + \sqrt{\rho\alpha}\right). \tag{108}$$

This means that, if $x_p$ is an $\alpha$-SOSP, it can be output by Algorithm 4. Therefore, Algorithm 4 will output a point with probability at least $1 - \frac{\omega'}{2}$.

Let $x_o$ be the output of Algorithm 4, for which we have

$$\|\nabla F(x_o)\|_2 \le \|\nabla F(x_o) - \nabla \bar{F}(x_o)\|_2 + \|\nabla \bar{F}(x_o)\|_2 \tag{109}$$

$$\le \|\nabla F(x_o) - \nabla f(x_o; \mathcal{S})\|_2 + \|\xi_o\|_2 + \|\nabla \bar{F}(x_o)\|_2, \tag{110}$$

and

$$\lambda_{\min}(\nabla^2 F(x_o)) \ge \lambda_{\min}(\nabla^2 F(x_o) - \nabla^2 \bar{F}(x_o)) + \lambda_{\min}(\nabla^2 \bar{F}(x_o)) \tag{111}$$

$$\ge -\|\nabla^2 F(x_o) - \nabla^2 \bar{F}(x_o)\|_2 + \lambda_{\min}(\nabla^2 \bar{F}(x_o)) \tag{112}$$

$$\ge -\|\nabla^2 F(x_o) - \nabla^2 f(x_o; \mathcal{S})\|_2 - \|H_o\|_2 + \lambda_{\min}(\nabla^2 \bar{F}(x_o)). \tag{113}$$

Following the same arguments for $x_p$, with probability at least $1 - \frac{\omega'}{2}$, we have the following $x_o$-related bounds hold:

$$\|\nabla F(x_o) - \nabla f(x_o; \mathcal{S})\|_2 \le O\left(\frac{G\log\left(\frac{d}{\omega'}\right)}{\sqrt{mn}}\right), \tag{114}$$

$$\|\xi_o\|_2 \le O\left(\frac{G\sqrt{dT\log\left(\frac{1}{\delta}\right)\log\left(\frac{1}{\omega'}\right)}}{\sqrt{m}n\epsilon}\right), \tag{115}$$

$$\left\|\nabla^2 f(x_o; \mathcal{S}) - \nabla^2 F(x_o)\right\|_2 \leq O\left(M\sqrt{\frac{\log\left(\frac{d}{\omega'}\right)}{mn}}\right), \tag{116}$$

$$\|\mathbf{H}_p\|_2 \leq O\left(\frac{Md\sqrt{T\log\frac{1}{\delta}\log\left(\frac{1}{\omega'}\right)}}{\sqrt{m}n\epsilon}\right). \tag{117}$$

Moreover, since $x_o$ is the output, we have

$$\|\nabla\bar{F}(x_o)\|_2 \leq \alpha + \frac{G\log\left(\frac{d}{\omega'}\right)}{\sqrt{mn}} + \frac{G\sqrt{dT\log\left(\frac{1}{\delta}\right)\log\left(\frac{1}{\omega'}\right)}}{\sqrt{m}n\epsilon}, \tag{118}$$

and

$$\lambda_{\min}\left(\nabla^2\bar{F}(x_o)\right) \geq -\left(\sqrt{\rho\alpha} + M\sqrt{\frac{\log\left(\frac{d}{\omega'}\right)}{mn}} + \frac{Md\sqrt{T\log\frac{1}{\delta}\log\left(\frac{1}{\omega'}\right)}}{\sqrt{m}n\epsilon}\right). \tag{119}$$

Combining these together, we finally obtain

$$\|\nabla F(x_o)\|_2 \leq O\left(\frac{G\log\left(\frac{d}{\omega'}\right)}{\sqrt{mn}} + \frac{G\sqrt{dT\log\left(\frac{1}{\delta}\right)\log\left(\frac{1}{\omega'}\right)}}{\sqrt{m}n\epsilon} + \alpha\right), \tag{120}$$

and

$$\lambda_{\min}(\nabla^2 F(x_o)) \geq -O\left(M\sqrt{\frac{\log\left(\frac{d}{\omega'}\right)}{mn}} + \frac{Md\sqrt{T\log\frac{1}{\delta}\log\left(\frac{1}{\omega'}\right)}}{\sqrt{m}n\epsilon} + \sqrt{\rho\alpha}\right). \tag{121}$$

Finally, by noting that $T = O\left(\frac{1}{\alpha^{2.5}}\right)$, and ignoring logarithmic factors and other irrelevant constant such as $G, M$, etc., we obtain $x_o$ is an $\alpha'$-SOSP for $\alpha'$ shown in the statement. By using the union bound, we have the two statements of Theorem 4 hold simultaneously with probability at least $1 - \omega'$. $\qquad\square$

