# OpenReview forum: "Private Stochastic Optimization for Achieving Second-Order Stationary Points"
_ICLR.cc/2025/Conference — Submitted to ICLR 2025_

### Official Review · Reviewer_vFr1 · 2024-11-03

**Soundness:** 3
**Presentation:** 3
**Contribution:** 2
**Rating:** 5
**Confidence:** 4

**Summary:**

The paper studies the convergence to a SOSP while preserving differential privacy. The paper points out an issue in the analysis of Liu et al. (2024) - incorrect handling of the stochastic noise - and shows how to fix it. It then presents a differentially private algorithm in the distributed settings.

**Strengths:**

A clear set of results on an interesting theoretically and well-motivated and studied problem.

**Weaknesses:**

Major concerns:

-- Please make it clear where you use non-constant learning rate. You don’t mention it in the main body and even at the crucial point - which seems to be Equation (41). This is a major concern since this is the part which, as you claim, is incorrect in Liu et al.

-- I believe the discussion “Gaps in Private SOSP selection” in Section 3 is not exactly fair.
1) First, one can compute the smallest Hessian eigenvalues efficiently - by using Hessian-vector products, which can be computed efficiently in neural networks.
2) Second, not sure what you mean by “requires additional data” - Liu et al. give bounds in Lemmas 4.1 and 4.5, and no additional data seems to be required.
3) Liu et al. didn’t try to guarantee privacy in the distributed settings, so I’m not sure one can call it a gap. That said, I’m not sure that even their approach has many concerns. First, one can approximate the gradient norm using sketches. Second, as outlined above, one don’t need to perturb Hessians, just the vectors resulting from the Hessian-vector products. Since this needs to be computed for at most logarithmic number of candidates, I expect the overall increase in the amount of noise to be negligible compared with the entire training procedure.

-- You want to find an α-SOSP for the population risk, while the proof seems to only cover the empirical risk (i.e. on the sampled dataset).

-- The choice of b1 and b2 depends on η and χ. On the other hand, η and χ depend on b1 and b2. Please eliminate the cyclic dependency, i.e. express b1 and b2 in terms of problem parameters. Please also express σ and r in the terms of problem parameters. Due to the cyclic dependency, combined with complicated expressions for b1 and b2, I couldn’t verify whether Theorem 2 is correct.

-- In Section 4, I don’t see why you can’t just use the analysis of Jin et al. black-box. The choice of parameters is the same, and the only change I see is how you check whether the point is a saddle point, by checking whether the point moved sufficiently far; this technique is, however, covered in their earlier paper “How to Escape Saddle Points Efficiently”.

**Questions:**

-- For all statements (at least for major theorems and lemmas), please:
1) State all assumptions and parameters precisely (you can state assumptions and refer to them later). To give an example, I spent quite some time looking for definition of U.
2) Refer to the algorithms they correspond to.
3) Refer to where they are proved in the appendix.

-- I personally believe the notation for the escaping radius is not great. I suggest using a more self-explanatory notation, such as R.

-- It’s not clear to me that the choice α = Θ(ψ) is the optimal choice. For example, in analysis of DP-SGD, alpha is basically independent of ψ (that is, you can make ψ and the number of iterations T arbitrarily large, without changing α), so it’s surprising to me that you have a hard dependence of α on ψ.

-- I think the paper would benefit from experimental evaluation.

-- You assume that each individual function (i.e. f(..., z)) is Lipschitz, smooth, and has Lipschitz Hessian. Do you need all of these assumptions for the individual functions?

---

> ### Author Response · Authors · 2024-11-20
> **Response to reviewer vFr1 (Part 1)**
>
> We thank the reviewer for dedicating valuable time to reviewing our paper and for the constructive feedback. We believe there are some misunderstandings that hinder a full comprehension of our work, and we are glad to provide clarifications.
>
> ## In the Weaknesses part:
>
> ### Weakness 1
>
> > "Please make it clear where you use a non-constant learning rate. You don’t mention it in the main body and even at the crucial point - which seems to be Equation (41). This is a major concern since this is the part which, as you claim, is incorrect in Liu et al."
>
> **Answer**: We explicitly specified the learning rate in Equation (4) of the main body. Unlike the learning rate $\eta = \frac{1}{M}$ used in (Liu et al., 2024), our learning rate is defined as $\eta = \frac{\sqrt{\rho\alpha}}{M^2 \iota^2}$. This setting introduces a dependency on the error rate $\alpha$, which further links the learning rate to critical parameters such as $\mathscr{T}$ and $\mathscr{S}$. These relationships are especially crucial in the analysis of our key Lemma~1, as can be seen in the derivations for Equations (39) and (41).
>
> ### Weakness 2
>
> > "I believe the discussion 'Gaps in Private SOSP selection' in Section 3 is not exactly fair.
> > 1. First, one can compute the smallest Hessian eigenvalues efficiently - by using Hessian-vector products, which can be computed efficiently in neural networks.
> > 2. Second, not sure what you mean by “requires additional data” - Liu et al. give bounds in Lemmas 4.1 and 4.5, and no additional data seems to be required.
> > 3. Liu et al. didn’t try to guarantee privacy in the distributed settings, so I’m not sure one can call it a gap. That said, I’m not sure that even their approach has many concerns. First, one can approximate the gradient norm using sketches. Second, as outlined above, one don’t need to perturb Hessians, just the vectors resulting from the Hessian-vector products. Since this needs to be computed for at most logarithmic number of candidates, I expect the overall increase in the amount of noise to be negligible compared with the entire training procedure."
>
> **Answer**: We address the reviewer's concern from the following 3 aspects:
> 1. **On Hessian eigenvalue computation**: While using the Hessian-vector product can reduce the scale of added noise, the additional error remains **non-negligible** compared to the training procedure. Specifically, perturbing the result vector instead of the full Hessian reduces the noise scale from $O(\frac{d\sqrt{T}}{\sqrt{m}n\epsilon})$ to $O(\frac{\sqrt{dT}}{\sqrt{m}n\epsilon})$. However, the $\sqrt{d}$ factor still imposes a significant challenge in high-dimensional settings. In contrast, in the single-machine case, the error in selecting DP-SOSP is independent of $d$ because noise is added directly to two one-dimensional quantities: the gradient norm and the minimum Hessian eigenvalue. This independence ensures that private model selection does not degrade the error guaranteed by training procedure. In distributed learning, however, perturbing vectors is unavoidable, leading to dimension-dependnet noise. **This fundamental limitation highlights the necessity of our proposed framework, which entirely eliminates the need for private model selection, thereby avoiding any additional computational and communication overhead.**
>
> 2. **On “additional data”**: By “additional data,” we refer to data beyond the training set that is reserved for private model selection, which results in accuracy degradation and incurs extra computational and communication costs in distributed settings. The bounds in Lemmas 4.1 and 4.5 of (Liu et al., 2024), as cited by the reviewer, are derived based on the number of data samples available for model selection, which are distinct from those used for training. This is evident in Theorem 4.6 of (Liu et al., 2024), where the total dataset of size $n$ is partitioned into two subsets: n/2 for training and n/2 for model selection, reducing effective size of the training set.
>
> 3. **On the word choice of “gap”**: We appreciate this suggestion. We think “limitation” is a more appropriate word than “gap” to describe this constraint. We have adopted this term in the revised manuscript for improved clarity and fairness.

---

> ### Author Response · Authors · 2024-11-21
> **Response to reviewer vFr1 (Part 2)**
>
> ### Weakness 3
>
> > "You want to find an $\alpha$-SOSP for the population risk, while the proof seems to only cover the empirical risk (i.e., on the sampled dataset)."
>
> **Answer**: We respectfully disagree with this assertion. Our proof addresses the population risk, as reflected in the unbiased stochastic gradient oracle of the population risk defined in Equation (3). Since the population risk is not directly observable, we estimate it (specifically, its gradients) using data samples—a standard approach in stochastic optimization supported by an extensive body of prior work, including (Choquette-Choo et al., 2024; Liu & Asi, 2024; Su et al., 2023; Su et al., 2022; Tao et al., 2022; Arora et al., 2023; Bassily et al., 2021), among others.
>
> Specifically, in line with closely related works (Liu et al., 2024; Jin et al., 2021), the sampling randomness in our gradient oracle is modeled using norm-sub-Gaussian random vectors, denoted as $\hat{\zeta}_t$’s. This general oracle ensures that all our subsequent arguments, from Lemma 1 through Lemma 6, are for the unknown population risk function $F$. Thus, our proof applies directly to the population risk.
>
> ### Weakness 4
>
> > “The choice of b1 and b2 depends on η and χ. On the other hand, η and χ depend on b1 and b2. Please eliminate the cyclic dependency, i.e. express b1 and b2 in terms of problem parameters. Please also express σ and r in the terms of problem parameters. Due to the cyclic dependency, combined with complicated expressions for b1 and b2, I couldn’t verify whether Theorem 2 is correct.”
>
> **Answer**: We believe there is a misunderstanding regarding the parameter dependencies. There is **no** cyclic dependency in the parameters. Both $\eta$ and $\chi$ ultimately depend on $\alpha$, as specified in Equation (4). The batch sizes $b_1$ and $b_2$ are also essentially determined by $\alpha$ through their relationships with $\eta$ and $\chi$, as outlined in Theorem 2. Theorem 2 expresses $\alpha$ explicitly in terms of problem parameters $n$, $d$, and $\epsilon$.
>
> We note that a similar exposition can also be observed in Lemma 3.6 and Theorem 4.6 of Liu et al. (2024). Specifically, their Lemma 3.6 establishes that $\alpha$ is determined by a high-probability gradient estimation error bound that depends on batch sizes $b_1$ and $b_2$, similarly to our Lemma 7. Then in their Theorem 4.6 (analogous to our Theorem 2), the set of $b_1$ and $b_2$ based on the pending $\alpha$ is used to finally drive the explicit form of $\alpha$.
>
> In response to the reviewer’s request, we provide explicit expressions for $b_1$ and $b_2$ as functions of problem parameters under $\widetilde{O}(\cdot)$ notation:
>
> $$
> b_1 = \\frac{n\\kappa}{2U\\eta} = \\widetilde{O}\\left(\\frac{n\\kappa M^2}{U\\sqrt{\\rho\\alpha}}\\right) \\\\
> = \\widetilde{O}\\left(\\frac{nM^{2} \\cdot \\max\\left\\{\\frac{G^{3/2}U^{1/2}\\rho^{1/2}}{M^{5/2}n^{1/2}}, \\frac{G^{14/15}d^{2/5}U^{4/5}\\rho^{8/15}}{M^{34/15}(n\\epsilon)^{4/5}}\\right\\}}{U\\sqrt{\\rho\\left(\\frac{1}{n^{1/3}} + \\left(\\frac{\\sqrt{d}}{n\\epsilon}\\right)^{2/5}\\right)}}\\right),
> $$
>
> $$b_2= \frac{n\eta\chi^2}{2U}=\widetilde{O}\left(\frac{n\sqrt{\rho}\alpha^{2.5}}{UM^2}\right)=\widetilde{O}\left(\frac{n\sqrt{\rho}\left(\frac{1}{n^{1/3}}+\left(\frac{\sqrt{d}}{n\epsilon}\right)^{2/5}\right)^{2.5}}{UM^{2}}\right).$$
> The upper bonds for $\sigma$ and $r$, provided in Lemma 7, can be similarly calculated via substituting above expressions for $b_1$ and $b_2$.

---

> ### Author Response · Authors · 2024-11-21
> **Response to reviewer vFr1 (Part 3)**
>
> ### Weakness 5
>
> > “In Section 4, I don’t see why you can’t just use the analysis of Jin et al. black-box. The choice of parameters is the same, and the only change I see is how you check whether the point is a saddle point, by checking whether the point moved sufficiently far; this technique is, however, covered in their earlier paper “How to Escape Saddle Points Efficiently”.
>
> **Answer**: We believe there is a misunderstanding.
>
> 1. Our parameter choice differ from that in (Jin et al., 2021), as clarified prior to Section~4.1. While their work focuses on iteration/gradient complexity to achieve any given target errorct $\alpha>0$, our work is constrained by differential privacy requirements, primarily the privacy budget $\epsilon$. These constraints inherently prevent $\alpha$ from being arbitrarily small, defining a subtly different problem that necessitates distinct parameter settings. **In particular, regarding the setting of $r$, in (Jin et al., 2021), $r$ is set based on the target $\alpha$, wheras in our work, $r$ is fully determined by privacy budget $\epsilon$.**
>
> 2. Our analysis differs fundamentally from (Jin et al., 2021) due to a critical difference in objectives. While they guarantee the *existence* of a second-order stationary point (SOSP) among all iterated models, our work aims to directly identify an SOSP. Achieving this goal requires a fundamentally different analysis of the dynamics of coupling sequences.
>
> Moreover, in (Jin et al., 2021), their Lemma B.3 for escaping saddle points relies on sufficient function value decrease during the escape. However, for stochastic optimization, the population risk function is unknown and cannot be directly evaluated, making function value decreases undetectable by the algorithm. Consequently, their approach can only guarantee the existence of an SOSP. In contrast, our work introduces a novel metric: model drift distance. We prove that a sufficiently large drift distance indicates a successful escape from saddle points, as demonstrated in Lemma 17. This drift is directly observable by the algorithm, allowing us to explicitly identify an SOSP. **As a result, our key lemma for saddle point escape (Lemma 1) fundamentally differs from Lemma B.3 in (Jin et al., 2021), employing a more granular analysis of coupling sequence dynamics that focuses on measurable model deviation rather than function value decreases.**
>
> 3. Regarding the paper “How to Escape Saddle Points Efficiently”, we note that it addresses a fundamentally different problem by focusing on perturbed gradient descent (GD) algorithms specifically applicable to empirical risk functions. These methods are not applicable to the stochastic optimization problem considered in our work. As previously mentioned, for empirical risk minimization, function value decreases can signal saddle point escape, as the function is directly observable. However, for population risk minimization, where the population risk function is unknown, this is infeasible.
>
> ## In the Question part:
>
> ### Question 1
>
> > “For all statements (at least for major theorems and lemmas), please:
> > 1. State all assumptions and parameters precisely (you can state assumptions and refer to them later). To give an example, I spent quite some time looking for definition of U.
> > 2. Refer to the algorithms they correspond to.
> > 3. Refer to where they are proved in the appendix.”
>
> **Answer**:
>
> 1. We agree that reorganizing Section 2 to centralize and explicitly state all assumptions and parameters would improve clarity. We have revised this section to ensure that all assumptions are grouped together for easier reference.
> 2. Regarding the correspondence of algorithms to theorems, we believe this has been addressed in our manuscript, as each theorem includes a reference to the relevant algorithm. However, if specific additional references are desired, we would greatly appreciate further clarification to address the concern.
> 3. We have added precise references to the proofs in the appendix for all major theorems and lemmas to streamline navigation, please see our revised manuscript.
>
> ### Question 2
>
> > “I personally believe the notation for the escaping radius is not great. I suggest using a more self-explanatory notation, such as R.”
>
> **Answer**: We appreciate the constructive feedback and will consider adopting a more self-explanatory notation in the final version to enhance clarity. However, for the current “author-reviewer” discussion phase, we prefer to retain the existing notation to avoid potential confusion with other reviewers.

---

> ### Author Response · Authors · 2024-11-21
> **Response to reviewer vFr1 (Part 4)**
>
> ### Question 3
>
> > "It’s not clear to me that the choice α = Θ(ψ) is the optimal choice. For example, in analysis of DP-SGD, alpha is basically independent of ψ (that is, you can make ψ and the number of iterations T arbitrarily large, without changing α), so it’s surprising to me that you have a hard dependence of α on ψ."
>
> **Answer**: The choice of $\alpha$ highlights a fundamental insight: the achievable second-order optimality is intrinsically linked to the magnitude of the stochastic gradient variance $\psi$. This relationship is also reflected in Lemma 3.6 of (Liu et al., 2024), where the error rate depends on the high-probability bound of the gradient estimation error. Specifically, in DP stochastic optimization for SOSP, $\alpha$ typically depends on both the gradient variance magnitude $\psi$ and the iteration number $T$, albeit with a logarithmic dependence on $T$.
>
> Given above, we believe there might be a misunderstanding in the reviewer's assertion. The dependence of $\alpha$ on $\psi$ is consistent with prior works and reflects the inherent difficulty of achieving sceond-order optimality under DP constraints.
>
> ### Question 4
>
> > "I think the paper would benefit from experimental evaluation."
>
> **Answer**: We thank the reviewer for this valuable suggestion. The primary focus of our paper is to address analytical issues in existing theoretical bounds. Given this focus, we believe empirical results would not provide additional insights into the analytical contributions presented in this work. Notably, previous studies in differentially private stochastic optimization, including those addressing FOSP (Arora et al., 2023) and SOSP (Liu et al., 2024), have also omitted empirical evaluations. We agree that an empirical study would be valuable to explore practical DP optimization applications, which we leave as future work.
>
> ### Question 5
> > "You assume that each individual function (i.e. f(..., z)) is Lipschitz, smooth, and has Lipschitz Hessian. Do you need all of these assumptions for the individual functions?"
>
> **Answer**: Yes, we require these assumptions. These assumptions align with those used in prior work on DP-SOSP, such as (Liu et al., 2024; Wang & Xu, 2021; Wang et al., 2019).

---

> ### Author Response · Authors · 2024-11-23
> **Follow-up with reviewer vFr1**
>
> Dear Reviewer vFr1,
>
> As the deadline approaches, we wanted to follow up to ensure that all your concerns regarding our paper have been addressed.
>
> If you believe we have adequately resolved your comments, we kindly hope you will consider re-evaluating our paper and adjusting your score accordingly.
>
> Thank you for your time and thoughtful feedback throughout this process.
>
> Many thanks,
>
> Authors

---

> ### Comment · Reviewer_vFr1 · 2024-11-26
>
> Thanks for your response, I have updated the score.

---

> > ### Author Response · Authors · 2024-11-26
> > **Follow-Up on Review Update**
> >
> > Dear vFr1,
> >
> > Thank you for taking the time to review our response and for updating your score. We sincerely appreciate your consideration and effort in evaluating our work.
> >
> > However, we noticed that the updated score still indicates a "marginally below the acceptance threshold" evaluation. We would be grateful if you could provide further clarification on any remaining concerns or aspects of our paper. Understanding your feedback would greatly help us address these points effectively and improve the work.
> >
> > Thank you again for your valuable input and time. We look forward to hearing from you.
> >
> > Best regards,
> >
> > Authors

---

### Official Review · Reviewer_mXDc · 2024-11-03

**Soundness:** 3
**Presentation:** 4
**Contribution:** 3
**Rating:** 8
**Confidence:** 4

**Summary:**

This paper studies a DP stochastic non-convex optimization problem with the focus on convergence to the second-order stationary points (SOSP), which is a new formulation and more stringent than previous focus first-order stationary points (FOSP). By revisiting the perturbed SGD with Gaussian noises, the paper corrects the error rate analysis in Liu et al. 2024 who ignored the statistic gradient variance, and improve the computational cost as well as the statistics error in the distributed learning setting in the previous literature Liu et al. 2024.

**Strengths:**

The paper is technically solid and well-written. The insight that model parameters move sufficiently far (over a threshold) leads to the novel algorithmic design as well as the theoretical analysis. The comparisons with the Liu et al. 2024 is solid and comprehensive. The extension to the distributed learning setting with improved computational cost and error guarantee showcase the power of the proposed algorithmic design and analysis.

**Weaknesses:**

A modest size of empirical experiments could help with the illustration of the practical value of the proposed novel algorithm.

**Questions:**

None.

---

> ### Author Response · Authors · 2024-11-21
> **Response to Reviewer mXDc**
>
> We sincerely thank the reviewer for the positive and supportive assessment of our work. We greatly appreciate your recognition of its contributions.
>
> Regarding the comment about experiments, the primary focus of our paper is to address analytical issues in existing theoretical bounds. As such, we believe empirical results would not provide additional insights into the analytical contributions of this work. Notably, prior studies in differentially private stochastic optimization, including those addressing FOSP (Arora et al., 2023) and SOSP (Liu et al., 2024), have similarly omitted empirical evaluations. However, we acknowledge that empirical studies would be valuable for exploring practical applications of DP optimization methods, and we leave it for future exploration.

---

### Official Review · Reviewer_vxfN · 2024-11-03

**Soundness:** 3
**Presentation:** 3
**Contribution:** 4
**Rating:** 8
**Confidence:** 3

**Summary:**

The paper presents a new framework for achieving SOSPs in DP non-convex optimization. The authors improved over the current state-of-the-art methods by better accounting over gradient variance and error rates based on a perturbed SGD . The framework is also extended to distributed environments.

**Strengths:**

The authors improve upon the state-of-the-art in the challenging area of private stochastic optimization for general non-convex functions.

The paper is a collection of elegant and technically solid results that advance the field through sophisticated analytical techniques.

The proposed algorithm represents a significant innovation compared to previous results.

**Weaknesses:**

For a reader who is not fluent with the literature, the paper is not easy to follow. Several insights are left unexplained. The paper is quite dense without enough provided intuition on the proposed algorithm. However, this is somehow understandable due to the page limit.

**Questions:**

Compared to "Private (Stochastic) Non-Convex Optimization Revisited: Second-Order Stationary Points and Excess Risks," the algorithm got rid of the internal state of "Frozen," which seems to be important. Could the authors further explain this insight?

The gap between the current result and the lower bound is still large (rate of n); any comments on this?

---

> ### Author Response · Authors · 2024-11-21
> **Response to Reviewer vxfN**
>
> We appreciate the reviewer’s recognition of our contributions, especially the novelty and technical sophistication of our algorithm. Below, we address the remaining questions.
>
> ### Question 1:
>
> > “Compared to "Private (Stochastic) Non-Convex Optimization Revisited: Second-Order Stationary Points and Excess Risks," the algorithm got rid of the internal state of "Frozen," which seems to be important. Could the authors further explain this insight?”
>
> **Answer**: Yes, this is indeed a significant improvement in our algorithm design. In (Liu et al., 2024), their Lemma 3.4 directly addresses saddle point escape by adding “extra” Gaussian noise to the DP gradient oracles whenever the gradient norm of the current point is small (indicating a potential saddle point). However, excessive “extra” Gaussian noise can degrade performance, so they introduce the internal state of “Frozen” to carefully manage this added noise.
>
> In contrast, our design and analysis fundamentally differ. Our approach leverages the Gaussian noise inherently present in the DP stochastic gradient oracles for escaping saddle points. This eliminates the need for adding separate “extra” Gaussian noise and, consequently, the internal state of “Frozen” becomes unnecessary in our algorithm.
>
> ### Question 2:
>
> > “The gap between the current result and the lower bound is still large (rate of n); any comments on this?”
>
> **Answer**:  We appreciate the reviewer’s sharp observation of the existing gap between our upper bound and the current DP lower bound. This gap remains an open question and represents a valuable direction for future research.
>
> On the upper bound side, whether our results can be further improved is an intriguing problem that requires in-depth exploration.
>
> On the lower bound side, we believe that part of the gap arises because the current DP lower bound, as stated in (Liu et al., 2024), is from Theorem 4.3 of Arora et al. (2023). This result was originally developed for *convex* loss functions and focuses on achieving *first-order* stationary points. Since finding a second-order stationary point is inherently more challenging than finding a first-order stationary point, we conjecture that the current lower bound may not be tight for the non-convex case. Establishing a tighter lower bound for achieving second-order stationary points in non-convex loss functions under DP constraints is a critical open problem that deserves further investigation.

---

> > ### Comment · Reviewer_vxfN · 2024-11-26
> >
> > Thank you for your response. I have no further comments.

---

> > > ### Author Response · Authors · 2024-11-26
> > > **Thanks for Your Support and Evaluation**
> > >
> > > Many thanks for taking the time to review our work. We truly appreciate your recognition and support, as well as your thoughtful evaluation throughout the process.

---

### Official Review · Reviewer_h4WK · 2024-11-04

**Soundness:** 3
**Presentation:** 3
**Contribution:** 4
**Rating:** 6
**Confidence:** 3

**Summary:**

This paper addresses the problem of achieving second-order stationary points (SOSP) in differentially private (DP) stochastic non-convex optimization. The authors identify two key limitations in the state-of-the-art Liu et al. (2024): (i) inaccurate error rates caused by the omission of gradient variance in saddle point escape analysis (ii) inefficiencies in private SOSP selection via the AboveThreshold algorithm.

The authors propose a new framework that leverages general gradient oracles to overcome these challenges.
The paper establishes a new DP algorithm that corrects existing error rates, and extends the approach to distributed settings.

**Strengths:**

- The paper is well structured and easy to follow. It clearly outlines the gap in state-of-the-art (SOTA) and the contributions.
- The paper points out an error in the analysis of SOTA and establishes the correct rate. The proposed DP-SPIDER seems simpler than the SOTA.
- The authors also consider the distributed settings and demonstrate the benefits of their proposed approach over the AboveThreshold used in SOTA.

**Weaknesses:**

The paper fixes the error in SOTA and establishes a correct rate. It would be nice if the authors could discuss if they think their results can be improved and how they compare with the results in SOTA if an easy fix is possible.

**Questions:**

Is the error in Liu et al. (2024) easy to fix? If so, what would be the correct error rate and how does it compare to your results?

---

> ### Author Response · Authors · 2024-11-21
> **Response to Reviewer h4WK**
>
> We sincerely thank the reviewer for recognizing the contributions of our work. We are pleased to address the reviewer’s remaining question below.
>
> ### Question:
>
> > “Is the error in Liu et al. (2024) easy to fix? If so, what would be the correct error rate and how does it compare to your results?”
>
> **Answer**: We interpret the reviewer’s question as asking whether the error in Liu et al. (2024) can be directly fixed through a revised proof for their algorithm. While this is feasible, such a correction would still fail to achieve the target SOSP with the optimal dependence on $\alpha$ required in our work: $\|\nabla F(x)\| \leq \alpha$ and $\nabla^2 F(x) \succeq -\sqrt{\rho \alpha} \cdot \mathbf{I}_d$. **Specifically, a direct correction would result in suboptimal second-order accuracy with a dependence of $\widetilde{O}(\alpha^{2/5})$, instead of the desired $\widetilde{O}(\alpha^{1/2})$.**
>
> The algorithm in Liu et al. (2024) can be viewed as a special single-machine case of the generic framework of perturbed gradient descent (GD) with bounded gradient inexactness, as developed in [YCRB19]. In this view, DP noise contributes to bounded gradient inexactness. The analysis in [YCRB19] implies the corrected convergence guarantees for the algorithm in (Liu et al., 2024). Assuming the first-order error rate satisfies $\|\nabla F(x)\| \leq O(\alpha)$, the analysis in [YCRB19] guarantees $\|\nabla F(x)\| \leq O(\alpha)$ and $\nabla^2 F(x) \succeq -\widetilde{O}(\sqrt{\rho} \alpha^{2/5}) \cdot \mathbf{I}_d$, see Theorem 3 of [YCRB19]. However, this falls short of the desired guarantee of $\nabla^2 F(x) \succeq -\widetilde{O}(\sqrt{\rho\alpha}) \cdot \mathbf{I}_d$, which is also what Liu et al. (2024) ideally aimed to achieve.
>
> Furthermore, Proposition 1 in [YCRB19] establishes a lower bound of $\widetilde{\Omega}(\alpha^{1/2})$ for dependence on $\alpha$ in second-order guarantees, highlighting the suboptimality of $\widetilde{O}(\alpha^{2/5})$. Additionally, Theorem 4 in [YCRB19] shows that with an exact gradient oracle, an optimal dependence of $\widetilde{O}(\alpha^{1/2})$ can be achievable. This explains why Liu et al. (2024) appeared to achieve the optimal order, as their analysis omitted the effect of gradient variance, as we discussed in Section 3 of our paper.
>
> In summary, while directly correcting the results of (Liu et al., 2024) using a refined analysis is feasible and can be accomplished with minimal effort based on [YCRB19], such corrections still cannot guarantee the target SOSP. **Designing a new framework, as we have done, is therefore both necessary and essential to meet these expectations.**
>
> We have incorporated this explanation into our paper to further strengthen the motivation for our work.
>
>     [YCRB19] Yin, D., Chen, Y., Kannan, R. &amp; Bartlett, P.. (2019). Defending Against Saddle Point Attack in Byzantine-Robust Distributed Learning. Proceedings of the 36th International Conference on Machine Learning (ICML 2019), in Proceedings of Machine Learning Research 97:7074-7084 Available from https://proceedings.mlr.press/v97/yin19a.html

---

> > ### Comment · Reviewer_h4WK · 2024-11-26
> >
> > Thank you for answering my questions.

---

> > > ### Author Response · Authors · 2024-11-26
> > > **Thanks for Your Positive Evaluation**
> > >
> > > We would like to thank you again for your positive evaluation of our paper.

---

### Author Response · Authors · 2024-11-25
**Kind Reminder: Your Feedback Matters – Please Check Our Response**

Dear Reviewers,

We sincerely appreciate your valuable time and effort in reviewing our paper and recognizing its contributions and quality. As the discussion phase is coming to an end, we kindly request you to review our responses and the revised manuscript. If you have any further questions, we would be more than happy to address them. We look forward to your kind re-evaluation of our work.

Thank you for your support!

Sincerely,

Authors

---

### Meta-Review · Area_Chair_M5ze · 2024-12-24

**Metareview:**

Paper studies the problem of privately achieving second order stationary point (SOSP). First, authors identify that current state-of-the-art (SOTA) method has an error in their theoretical analysis and its SOSP selection sub-routine is inefficient. Then they provide a new algorithm and analysis techniques which sidesteps this inefficiency and provide new SOTA DP optimization results. Paper seems to contain correct theory and interesting techniques even though it is dense and hard to parse.

Authors do not provide any empirical evaluation and justify their choice to do so by appealing to standards set by two recent works on this topic. Empirical evaluation is considered important in both ML and Optimization algorithms literature, and particularly in the ICLR conference. For example, proper experimental evaluation of the claims in the prior SOTA work may have identified the error in the analysis. Therefore, the authors are strongly encouraged to add experiments in their next revision.

PS: Notations used in the paper were challenging to parse, especially the non-standard calligraphic letter which all looked similar.

**Additional Comments On Reviewer Discussion:**

Authors addressed most of the questions around clarity, high-level explanations, novelty, and limitations of prior work. However, authors did not address the lack of empirical evaluation.

---

### Decision · Program_Chairs · 2025-01-22

Reject